

# Effects of transport on a biomass burning plume from Indochina

# during EMeRGe-Asia identified by WRF-Chem

**Chuan-Yao Lin[1]\*, Wan-Chin Chen[1], Yi-Yun Chien[1], Charles C. K. Chou[1], Chian-Yi Liu[1], Helmut Ziereis[2], Hans Schlager[2], Eric Förster[3], Florian Obersteiner[3], Ovid O. Krüger[4], Bruna A. Holanda[4], Mira L. Pöhlker[4,a], Katharina Kaiser[5,7], Johannes Schneider[5], Birger Bohn[8] , Maria Dolores Andrés Hernández[6], John P. Burrows[6]**

1. Research Center for Environmental Changes, Academia Sinica, Taipei, Taiwan
2. Deutsches Zentrum für Luft- und Raumfahrt (DLR), Institut für Physik der Atmosphäre, Oberpfaffenhofen, Germany
3. Karlsruhe Institute of Technology, Institute of Meteorology and Climate Research, Karlsruhe, Germany
4. Multiphase Chemistry Department, Max Planck Institute for Chemistry, Mainz, Germany
5. Particle Chemistry Department, Max Planck Institute for Chemistry, Mainz, Germany
6. Institute of Environmental Physics, University Bremen, Bremen, Germany
7. Institute for Atmospheric Physics, Johannes Gutenberg University, Mainz, Germany
8. Institute of Energy and Climate Research IEK-8, Forschungszentrum Jülich, Jülich, Germany

[a]now at: Faculty of Physics and Earth Sciences,Leipzig Institute for Meteorology, University of Leipzig/Experimental Aerosol and Cloud Microphysics Department, Leibniz Institute for Tropospheric Research, Leipzig, Germany

\*Corresponding author
**Chuan Yao Lin,**
Research Center for Environmental Changes, Academia Sinica, Taipei, Taiwan
128 Sec. 2, Academia Rd, Nankang, Taipei 115, Taiwan
(E-mail: yao435@rcec.sinica.edu.tw, Tel.: +886-2-27875892, Fax: +886-2-27833584),

**Abstract.**



The Indochina biomass burning (BB) season in springtime has a substantial
environmental impact on the surrounding areas in Asia. In this study, we evaluated the
environmental impact of a major long-range BB transport event on 19 March 2018 (a
flight of the HALO research aircraft, flight F0319) preceded by a minor event on 17
March 2018 (flight F0317). Aircraft data obtained during the campaign in Asia of the
Effect of Megacities on the transport and transformation of pollutants on the Regional
to Global scales (EMeRGe) were available between 12 March and 7 April 2018. In the
F0319, results of 1-min mean carbon monoxide (CO), ozone ($O_3$), acetone (ACE),
acetonitrile (ACN), organic aerosol (OA) and black carbon aerosol (BC) concentrations
were up to 312.0 ppb, 79.0 ppb, 3.0 ppb, 0.6 ppb, 6.4 µg m$^{-3}$, 2.5 µg m$^{-3}$ respectively,
during the flight, which passed through the BB plume transport layer (BPTL) between
the elevation of 2000–4000 m over the East China Sea (ECS). During F0319, CO, $O_3$,
ACE, ACN, OA and BC maximum of the 1 minute average concentrations were higher
in the BPTL by 109.0 ppb, 8.0 ppb, 1.0 ppb, 0.3 ppb, 3.0 µg m$^{-3}$ and 1.3 µg m$^{-3}$
compared to flight F0317, respectively. Sulfate aerosol, rather than OA, showed the
highest concentration at low altitudes (<1000 m) in both flights F0317 and F0319
resulting from the continental outflow in the ECS.
The transport of BB aerosols from Indochina and its impacts on the downstream
area was evaluated using a WRF-Chem model. Over the ECS, the simulated BB
contribution demonstrated an increasing trend from the lowest values on 17 March 2018
to the highest values on 18 and 19 March 2018 for CO, fine particulate matter ($PM_{2.5}$),
OA, BC, hydroxyl radicals (OH), nitrogen oxides ($NO_x$), total reactive nitrogen ($NO_y$),
and $O_3$; by contrast, the variation of $J(O^1D)$ decreased as the BB plume's contribution
increased over the ECS. In the low boundary layer (<1000 m), the BB plume's
contribution to most species in the remote downstream areas was <20 %. However, at
the BPTL, the contribution of the long-range transported BB plume was as high as 30–



80 % for most of the species ($NO_y$, $NO_x$, $PM_{2.5}$, BC, OH, $O_3$, and CO) over South China
(SC), Taiwan, and the ECS. BB aerosols were identified as a potential source of cloud
condensation nuclei, and the simulation results indicated that the transported BB plume
had an effect on cloud water formation over SC and the ECS on 19 March 2018. The
combination of BB aerosol enhancement with cloud water resulted in a reduction of
incoming shortwave radiation at the surface in SC and the ECS which potentially has
significant regional climate implications.


**1 Introduction**
Biomass burning (BB) is one of the main sources of aerosols, greenhouse gases, and air
pollutants (e.g. Ramanathan et al., 2007; Lin et al., 2009; 2014; Tang, 2003; Carmichael
et al., 2003; Chi et al., 2010; Fu et al., 2012; Lin N.H. et al., 2012; Chuang et al., 2016).
Reid et al. (2013) and Giglio et al. (2013) investigated the seasonal aerosol optical depth
over Southeast Asia and have indicated that Indochina is a major contributor of carbon
emissions in springtime. Galanter et al. (2000) estimated that BB accounts for 15–30 %
of the entire tropospheric CO background. Huang et al. (2013) indicated that the
contribution of BB in Southeast Asia to the aerosol optical depth (AOD) in Hong Kong
and Taiwan could be in the range of 26-62 %. Moreover, BB emissions over Indochina
are a significant contributor to black carbon (BC), organic carbon (OC), and $O_3$ in East
Asia (Lin et al., 2014). In their BB modeling study, Lin et al. (2014) identified a
northeast (NE) to southwest (SW) zone stretching from South China (SC) to Taiwan
with a reduction in shortwave radiation of approximately 20 W $m^{-2}$ at the ground
surface. In addition, the total carbon emission from BB in Southeast Asia is
approximately 91 Tg C $yr^{-1}$, accounting for 4.9 % of the global total (Yadav et al.,
2017). According to Xu et al. (2018), BB in Indochina leads to BC production at high



concentrations of up to 2–6 µg m$^{-3}$ in spring. The authors reported that BC particles
were transported to the glaciers in the Tibetan Plateau, where it significantly affected
the melting of the snow, causing some severe environmental problems, such as water
resource depletion. Ding et al. (2021) indicated that BB aloft aerosols strongly increase
the low cloud coverage over both land and ocean and affect the monsoon in the
subtropical Southeast Asia.
Although many researchers have indicated the importance of BB emissions, their
precise estimation and applying in the modeling study remains challenging (Fu et al.
2012; Huang et al. 2013; Pimonstree et al. 2018; Marvin et al. 2021). For example,
Heald et al. (2003) conducted an emission inventory in Southeast Asia and reported that
the uncertainties of BB emission estimations could be a factor of three or even higher.
Following an inverse model analysis, Palmer et al. (2003) also indicated the
overestimation of regional BB emissions over Indochina. Shi and Yamaguchi (2014)
pointed out BB emissions exhibited similar temporal trends between 2001 and 2010
and with strong interannual variability over southeast Asia. Satellite data can be used
to easily locate hotspots such as those where agricultural residuals burning and forest
wildfires are occurring worldwide. However, accurately quantifying the amount of BB
emission from satellite data is difficult because anthropogenic pollutants and BB
emissions are typically mixed in the atmosphere. During the NASA Transport and
Chemical Evolution over the Pacific (TRACE-P) aircraft mission in spring 2001, Jacob
et al. (2003) observed that warm conveyor belts (WCBs) lift both anthropogenic and
BB (from SE Asia) air pollution to the free troposphere, resulting in complex chemical
signatures. Wiedinmyer et al. (2011) demonstrated that the uncertainty of emission
estimation could be as high as a factor of 2 because of the error introduced by estimates
in fire hotspots, area burned, land cover maps, biomass consumption, and emission
factors in the model. In this context, Lin et al. (2014) highlighted the uncertainty of



emission estimation in the first version of Fire Inventory from NCAR (Wiedinmyer et
al., 2011).

The transport of BB pollution is strongly dependent on the atmospheric structure

and weather conditions. Tang et al. (2003) noted that most BB aerosols, having their
source in Indochina (mainly south of 25 °N and be alofted to an altitude of 2000–4000
m) during the TRACE-P campaign were associated with outflow in the WCB region
after frontal passage. Lin et al. (2009) suggested a mountain lee-side troughs as an
important mechanism, resulting in BB product transport from the surface to >3000 m.
BB pollution is often transported from its sources to the East China Sea (ECS), Taiwan,
and the western North Pacific within a few days.

The airborne field experiment EMeRGe ( Effect of Megacities on the transport and

transformation of pollutants on the Regional to Global scales) over Asia was led by the
University of Bremen, Germany and conducted in collaboration with Academia Sinica,
during    the    inter-monsoon    period    in    2018    (http://www.iup.uni-
bremen.de/emerge/home/home.html). The EMeRGe aircraft mission consists of two
parts. The first mission phase was conducted in Germany in July 2017 and the second
phase was conducted from Taiwan in 2018 (Andrés Hernández et al. 2022).EMeRGe in
Asia aimed at the investigation of the long range transport (LRT) of local and regional
pollution originating in Asian major population centers (MPCs) from the Asian
continent into the Pacific. A central part of the project was the airborne measurement
of pollution plumes on-board of the High Altitude and Long Range Research Aircraft
(HALO). The HALO platform was based in Tainan, Taiwan (Fig. 1a-b), and made
optimized transects and vertical profiling in regions north or south of Taiwan,
dependent on the relevant weather and emission conditions. HALO measurements
additionally provide important information for the evaluation of the LRT of BB
emissions and its potential environmental impact in East Asia between 12 March and 7


April 2018. During the EMeRGe-Asia campaign, HALO carried out 12 mission flights
in Asia and 4 transfer flights from Europe to Asia with a total of 110 flight hours.
This paper is organized as follows: the model configuration and BB emission
analysis employed in the model simulation are described in Section 2, and the weather
conditions and HALO measurement results are presented in Section 3. The model
performance, as well as the evaluation of BB product transport and effects on East Asia
selected regions are discussed in Sections 4 and 5, respectively.

**2 Aircraft data and Model configuration**
**2.1 HALO aircraft data**
The HALO aircraft was equipped with a number of instruments and a detailed
description of the measurement systems onboard the HALO was presented in Andrés
Hernández et al.(2022). In this study, aerosol data (OA, BC, $SO_4^{2-}$, $NO_3^-$, $NH_4^+$), and
trace gases such as CO, $SO_2$, $O_3$, $NO_x$, $NO_y$, acetone (ACE), acetonitrile (ACN), HCHO,
HONO, OH, $HO_2$, and photolysis rate $J(O^1D)$, $J(NO_2)$ were employed in the analysis.
**2.2   WRF-Chem Model and model configuration**
We used the Weather Research Forecasting with Chemistry (WRF-Chem) model (Ver.
4.1.1) (Grell et al., 2005) to study the LRT of air masses associated with BB pollutants
in Indochina. The initial and boundary meteorological conditions for WRF-Chem were
obtained from National Centers for Environmental Prediction (NCEP)-GDAS Global
Analysis data sets at 6-h intervals. The Mellor–Yamada–Janjic planetary boundary
layer scheme (Janjic, 1994) was applied. The horizontal resolution for the simulations
performed was 10 km, and the grid box had $442 \times 391$ points in the east–west and
north–south directions (Fig. 1a). A total of 41 vertical levels were included, with the
lowest level at an elevation of approximately 50 m. To improve the accuracy of the
meteorological fields, a grid nudging four-dimensional data assimilation scheme was



applied using the NCEP-GDAS Global Analysis data.

The cloud microphysics used followed the Lin scheme (Morrison et al., 2005). The

rapid radiative transfer model (Zhao et al., 2011) was used for both longwave and
shortwave radiation schemes. Moreover, land surface processes are simulated using the
Noah-LSM scheme (Hong et al., 2009). In terms of transport processes, we considered
advection by winds, convection by clouds, and diffusion by turbulent mixing. The
removal processes in this study were gravitational settling, surface deposition, and wet
deposition (scavenging in convective updrafts and rainout or washout in large-scale
precipitation). The kinetic preprocessor (KPP) interface was used in both of the
chemistry schemes of the Regional Atmospheric Chemistry Mechanism (RACM,
Stockwell et al., 1990). The secondary organic aerosol formation module, the Modal
Aerosol Dynamics Model for Europe (Ackermann et al., 1998)/Volatility Basis Set
(Ahmadov et al., 2012), was also employed in the WRF-Chem model. In RACM, "KET"
is the only species available for ketones. Thus, we do estimate the measurement of ACE
by using simulated KET in this study.

**2.3 Emission Inventories**
Anthropogenic emissions, such as $NO_x$, CO, $SO_2$, nonmethane volatile organic
compounds, sulfate, nitrate, $PM_{10}$, and $PM_{2.5}$, were adopted on the basis of the emission
inventory in Asia – MICS-Asia III (Li et al., 2020; Kong et al., 2020). For BB emissions
FINNv1.5 (https://www.acom.ucar.edu/Data/fire/) was employed. FINN provided
daily, 1000 m resolution, global estimates of the trace gas and particle emissions from
open BB, which included wildfires, agricultural fires, and prescribed burning but not
biofuel use and trash burning (Wiedinmyer et al., 2011). The anthropogenic emissions
in Taiwan were obtained from the Taiwan Emission Data System (TEDS) which is the
emission inventory of the air-pollutant monitoring database of the Taiwan





Environmental Protection Administration. The TEDS version used for this study was
V9.0 (2013) and contained data on eight primary atmospheric pollutants: CO, NO, $NO_2$,
$NO_x$, $O_3$, $PM_{10}$, $PM_{2.5}$, and $SO_2$.

**3 Characteristics of the field experiment**
**3.1 MODIS Aerosol optical depth and Weather conditions**

Figures 2a and b visualizes the numerous fire hotspots and high aerosol optical depth

on 17 March 2018 registered by the MODIS satellite. Indeed, a large number of BB fire
hotspots frequently occurred over Indochina during the springtime (Lin et al. 2009;
2014) and EMeRGe-Asia campaign (supplementary Figure S1). On 17 March 2018 at
06:00 UTC (14:00 LT; LT = UTC+8:00) the weather data indicated a series of high-
pressure systems in northern China and a separate high-pressure system over Korea
(Fig. 2c). At 1000 hPa, a strong northerly continental outflow was identified over
southern Japan, the ECS, and Taiwan (Fig. 2d). On 19 March 2018, a new frontal
system was located from Korea to the Guangdong province in SC (Fig. 2e). On the
same day at 06:00 UTC, a discontinued flow was identified at the frontal zone to the
north of Taiwan in the ECS (Fig. 2f). In other words, Taiwan was located at the
prefrontal and warm conveyor area due to the surrounding southerly flow on 19 March
2018 at 06:00 UTC (Figs. 2e and 2f, respectively). The southerly wind was gradually
replaced by the northeasterly after another frontal passage on 20 March 2018 at 00:00
UTC (data not shown).

In the upper layer (700 hPa; Figs. 2g–2j), the flow pattern differed from that at the

near-ground surface (1000 hPa; Figs. 2d and 2f). A southwesterly strong wind, coming
from the east side of the Tibetan Plateau in SC, moving to the North Eats i.e. Korea, is
converted to a polar front wave flow in northeastern China and Korea on 17 March
2018 (Fig. 2g). This high-elevation northward strong wind belt distribution at 700 hPa



was associated with a corresponding lee-side trough at the east of the Tibetan Plateau,
whereas a ridge was noted over the east coast of China on the same day (Fig. 2h).
Consistent with the mechanism reported by Lin et al. (2009), once a significant lee-side
trough formed, it provided favorable conditions for the upward motion over the lee-side
of the Tibetan Plateau and brought BB emission to the free troposphere layer following
the strong wind belt transport to the downwind area. After the weather system moved
to the east, the north–south trough turned to SW–NE such that the strong wind belt was
in an approximately SW–NE direction and located between 20 and 30 °N on 19 March
2018 (Figs. 2i and 2j). In conclusion, the Indochina BB pollutants were driven by the
strong wind belt from Indochina, northward to SC on 17 March 2018 and then eastward
passing over Taiwan between 20 and 30 °N to the south of Japan on 19 March 2018.
**3.2   Characteristics of LRT BB to the ECS by WRF-Chem model**

Figure 3 shows latitude longitude plots of the simulated CO concentration

differences with and without BB emission at an elevation of 1000 m (Fig. 3a), mainly
in Indochina, SC, and the South China Sea on 17 March 2018. The ambient flow was
easterly and then northward from the South China Sea to SC at 1000 m elevation
between 00:00 and 12:00 UTC on 17 March 2018 (Fig. 3a-b). The BB plume
accumulated and persisted for an extended period in the lower part of the boundary
layer on 17 and 19 March 2018 (Figs. 3a-b, and 3e-f). In contrast, the high CO
concentration followed the southwesterly or westerly strong wind belt (Figs. 3c-d, and
3g-h) and its weather conditions (Fig. 2) at an elevation of 3000-m (700 hPa). Following
the movement of the ridge and trough at the 700 hPa geopotential height (Fig. 2h and
2j), the associated strong wind belt turned to move eastward in the SW–NE direction
between 17 and 19 March 2018. The BB plume transport over Indochina was affected
by a fast-moving strong flow at 700 hPa (Fig. 2g and 2i), shifting the plume toward
Taiwan and the ECS, during 17–19 March 2018. The highest CO concentration
contributed by the BB plume was >150 ppb, originally sourced from Indochina, and it
was mainly transported northward on 17 March 2018 (Figs. 3c-d) and then covered a
large area in East Asia at a CO concentration of >100 ppb on 19 March 2018 (Figs. 3g-
h). Figure 4 indicates simulation differences for the contribution of BB along an E–W
cross-section at 30 °N at 16:00 UTC on 18 March 2018 (Fig. 4a) and 06:00 UTC on 19
March 2018 (Fig. 4b). We noted that a strong wind at 2000 m elevation and a high CO
concentration (>70 ppb) due to BB at the BPTL. Moreover, the CO concentration
attributed to BB was low at the elevation of >4000 m on 19 March at 06:00 UTC (Fig.
4b), showing that the BB pollutants mainly affect altitudes below 4000 m.
**3.3 Aircraft measurements**
Two HALO flights were scheduled to the ECS to measure the pollutants following the
continental outflow; the flights departed on 17 (Fig. 5a) and 19 (Fig. 6a) March 2018
and followed similar tracks. To indicate the measurement results along the flight path,
the 1-min average data is shown in Figures 5b and 6b.   On 17 March 2018, the flight
departed from Tainan (Fig. 1b) at 01:09 UTC (09:09 LT) first southbound and then
northward to the ECS (Fig. 5a). The elevation for sample collection was mainly <4000
m, where the CO concentration was found to be <200 ppb in most cases on that day
(Fig. 5b). At elevations between 2000 and 4000 m, the concentration of the major
aerosol components (i.e., OA, BC, $SO_4^{2-}$, $NO_3^-$, and $NH_4^+$ ) was mostly <2 µg m$^{-3}$,
except just above western Taiwan after 08:00 UTC (Figs. 5a–5d). The peak
concentrations for OA, BC, $SO_4^{2-}$, $NH_4^+$, and $NO_3^-$ were 3.4, 1.2, 2.1, and 0.7 µg m$^{-3}$,
respectively, at the altitude between 2000 and 4000 m.   $SO_4^{2-}$ demonstrated the highest
concentration among the aerosol components, especially during 04:00–04:37 and
05:48–06:15 UTC (peaking at 5.1 µg m$^{-3}$) when the flight was north of 30 °N and an
elevation of <1000 m (Figs. 5a–5c). This result could be attributed to anthropogenic
pollution from the continental outflow (Lin et al. 2012) or probably part from Japan





contributed to the high sulfate concentration in the boundary layer over the ECS. As for
the trace gases such as ACE, ACN and $O_3$, their concentrations between 2000 and 4000
m were stable and ranged between 1-2 ppb, 0.1-0.3 ppb, and 60-70 ppb (Fig. 5b),
respectively, implying minor influence over the ECS by the BB plume in this flight.
Figure 5e illustrates the HYSPLIT (Stein et al., 2021) 96-h backward trajectories, which
identified the air mass origin starting at 02:00 UTC, followed by 04:00, 06:00, and
09:00 UTC. The continental outflow contributed to higher sulfate concentrations (3–5
$\mu g\ m^{-3}$ at 33 °N) at 04:00 and 06:00 UTC (Figs. 5b, 5c, and 5e) at <1000 m along the
flight path. In contrast, south of 25 °N and above Taiwan, the local pollution and
continental outflow are dominating sources on 17 March 2018.
The HALO flight on 19 March 2018 departed at 00:19 UTC (08:19 LT). It was
bound northward and sampled air at an altitude of <4000 m most of the time, as shown
in Figures 6a and 6b. Figures 6c and 6d indicate the latitude-height variation of $SO_4^{2-}$
and OA mass concentrations along the flight path on 19 March 2018. As the flight left
Taiwan, it maintained an elevation of 3000 m during 01:00–02:00 UTC (Fig. 6a, 121–
126 °E) and then descended to <1000 m during 02:00–02:40 UTC (Fig. 6b). The OA
mass concentration was higher at 3000 m than at the low altitude during 01:00–03:00
UTC (Figs. 6b and 6d). In particular, CO, OA and BC exhibited a substantial peak
concentration of 312 ppb, 6.4 $\mu g\ m^{-3}$ and 2.5 $\mu g\ m^{-3}$ at 01:54 and 02:51 UTC at 26 °N,
125–126 °E, and an altitude of 2000–4000 m, where a BPTL was observed. The trace
gases such as ACE, ACN, and even $O_3$ (Fig. 6b) have consistent peak times in the BPTL
with concentrations of 3.0 ppb, 0.6 ppb, and 79 ppb, respectively. In this flight, $SO_4^{2-}$
had the second-highest concentration among the aerosol components (1–2.4 $\mu g\ m^{-3}$;
Figs. 6b and 6c) upstream of Taiwan (25–27 °N) during 1:00–3:00 UTC.
In the northern part of the flight between 03:00 and 05:00 UTC at an elevation of
>3000 m, the aerosol component concentrations were all at their lowest level (Figs. 6b–


6d). During 05:00–07:00 UTC, the HALO aircraft flew back southward to 25 °N, where
high OA mass concentrations appeared again between 2000 and 4000 m (Figs. 6a, 6b,
and 6d). Sulfate was the species with the highest concentration between 05:30 and
06:30 UTC (Figs. 6b and 6c) when the flight's elevation was <1000 m in the lower
boundary between 25 and 27 °N (upstream of Taiwan). The reason explaining this
observation is that the transport of anthropogenic pollutants of continental origin takes
place mainly in the boundary layer (Figs. 6b–6d). Other aerosol species, such as $NO_3^-$
and $NH_4^+$, demonstrated low concentrations, except when the elevation was <1000 m,
where they ranged up to 1 µg m$^{-3}$ (Fig. 6b).

The 96-h HYSPLIT backward trajectory starting from the flight locations at

02:00–07:00 UTC (Fig. 6e) indicated that the air masses at elevations between 2000
and 4000 m were potentially transported from Indochina. North of 30 °N and at altitudes
of >3000 m at 04:00 UTC, the concentrations of air pollutants (including OA, $SO_4^{2-}$,
$NO_3^-$, and $NH_4^+$) were low (Figs. 6b and 6e) even though the air mass in the low
boundary was sourced from SC and the Taiwan Strait. In general, the results are
consistent with those of Lin et al. (2009, 2014), Carmical et al. (2003), and Tang et al.
(2003): the BPTL was mainly located south of 30 °N. The fact that higher OA was
observed rather in the higher altitudes than in the lower boundary also demonstrated
the vertical distribution over the ECS.

Figure 7 displays the vertical distribution of the gases and major aerosol

components found on the flights on 17 (blue) and 19 (green) March 2018 as well as the
mean concentrations noted in the seven flights (on 17, 19, 22, 24, 26, and 30 March and
4 April 2018; red) to the ECS during EMeRGe-Asia. Figure 7 illustrates all profiles
calculated as 1-min mean and every 500-m interval with one standard deviation (±σ).
The number of the data points is displayed on the right side of each figure. The mean
CO concentration profile demonstrated a decreasing trend from 240 ppb near the



ground to 150 ppb at an altitude of 2500 m and 140–160 ppb at altitudes >6000 m (Fig.
7a). The concentration for 17 March 2018 (flight F0317) was similar to the mean
concentration profile, except for that at the <1500 m elevation in the lower boundary.
However, a higher CO concentration (40–80 ppb) enhancement was noted on 19 March
2018 (flight F0319) than the mean profile and flight F0317. The mean difference in CO
concentration between flights F0319 and F0317 was as high as 80 ppb at an elevation
of 3000-3500 m (Fig. 7a). Similarly, OA concentration was significantly higher in the
BPTL vertical distribution in flight F0319 than in the mean profile and flight F0317
(Fig. 7b). The mean OA concentration for the flight F0319 peaked at an elevation of
2000–2500 m, increasing to 2 $\mu g\ m^{-3}$ more than in the mean profile and flight F0317.
Other aerosol components such as $SO_4^{2-}$, $NH_4^+$, and $NO_3^-$ (Supplementary Fig. S2a-c)
also had a similar vertical distribution trend, but the concentration differences were
minor compared with OA concentrations. The magnitude of the maximum differences
between the flights F0319 and F0317 in the BPTL was 1.3, 0.7, and 0.4 $\mu g\ m^{-3}$ for
$SO_4^{2-}$, $NH_4^+$, and $NO_3^-$, respectively. The maximum difference concentration of BC can
be as high as 1.2 $\mu g\ m^{-3}$ at 2000-2500 m between the flights F0319 and F0317 (Fig.7c).
Regarding the variation in hydrocarbon species such as ACN (Fig. 7d) and ACE ( Fig.
7e) in the BPTL, their maximum mean concentrations in the flight F0319 were higher
than those in the profile of the flight F0317 by 0.18 and 0.9 ppb, respectively. In other
words, flight F0319 had a more significant impact on the CO, OA, BC, and volatile
organic compound (VOC) species such as ACN and ACE in the BPTL, which might
account for the effect of BB emission transport from Indochina. The ozone
concentration was lower in both flights F0317 and F0319 than in the mean profile at
the elevations <2000 m (Fig. 7f). The ozone titration by $NO_x$ in the low boundary might
also play a role. However, it was approximately 5–7 ppb higher in the flight F0319 than
in the flight F0317 between the elevations of 1500 and 3000 m. In their downwind area,





LRT of BB emissions might increase this concentration further at the BPTL (Tang et
al., 2003; Lin et al., 2014) and also discussed in section 4. By contrast, the J value
[J(O$_1$D)] (Fig. 7g) was higher for flight F0317 than for F0319 in the elevation range 1000–
3000 m, in line with high aerosol concentrations and associated cloud enhancement that
typically lead to decreased photolysis frequencies [i.e., J(O$_1$D)] (Tang et al., 2003).
Consistently, at altitudes >4000 m the presence of clouds below the aircraft led to greater J
values. The concentrations of other species such as NO$_y$ (Fig.7h) and HONO
(Supplementary Fig. S2d) were also greater in flight F0317 than in flight F0319 by 0.4-
1.2 ppb and 10-34 ppt, respectively, in the low boundary (<1500 m). At the BPTL, the
concentration of NOy (1-2 ppb) in the flight F0319 was higher than in the flight F0317,
but the difference was less than 0.6 ppb. The results from the TRACE-P campaign,
which examined the Asian outflow of NO$_y$, also demonstrated large increases in NO$_y$
concentrations (0.5-1 ppb) downwind from Asia. The NO$_y$ consisted mainly of HNO$_3$
and peroxyacetyl nitrate (Miyazaki et al., 2003; Talbot et al., 2003).

**4 Simulation results and discussion**
**4.1 Model performance and BB transport identification**
Tables 1 and 2 and Fig. 8 plot the Pearson correlation coefficients between 5-min
merged observations on board the HALO and the simulation for flights F0317 and
F0319. Meteorological parameters such as potential temperature (theta), relative
humidity (RH), and wind speed (WS) were all captured well by the model along the
HALO flight path on the 2 days. The correlation coefficient (R) for meteorological
parameters was high, ranging from 0.92 to 0.99 (Table 1). The strong correlation
indicates the high representativeness of the reanalysis of meteorological data used in
the simulation. Among the trace species and aerosol components, toluene (TOL), NO$_x$,
ketones (KET), BC, OA, HONO, SO$_2$, and HCHO demonstrated an R of >0.5 (good





correlation) and CO and O₃ showed an R of nearly 0.5 (Table 1). The simulation
performance was investigated in the BL (<1000 m; Fig. 8), at 2000–4000 m altitude
(Table 2 and Fig. 8) and for the whole period of both flights (Table 1 and Fig. 8; blue
dot). Even in the BPTL, the simulated meteorological parameters presented a good
correlation (R > 0.93), followed by KET, OA, BC and CO (R > 0.6) as well as O₃ and
NO$_y$ (R > 0.5) (Table 2). In other words, at the BPTL, the R for the simulation
significantly increased for OA, BC, CO, and KET (Tables 1 and 2 and Fig. 8), which
are indicators for BB being a source of pollution in the model. In contrast, $SO_4^{2-}$, $NO_3^-$,
$NH_4^+$, $SO_2$, $NO_2$, and HCHO had better correlation in the lower part of the boundary
layer, at altitudes <1000 m (see Fig. 8) than in the BPTL. We explain this by the
transport of anthropogenic pollutants in the continental outflow in the lower part of the
boundary layer in ECS.

The modeling results tended to overestimate the concentration of the species, with

examples being CO (59 ppb), OA (0.5 µg m$^{-3}$), BC (0.3 µg m$^{-3}$) and O₃ (12.1 ppb;
Table 2) in the BPTL. Because high concentrations of CO, BC and OA in BPTL are
accurate indicators of BB in the model, the BB emission from the source of FINN data
are probably also overestimated (Lin et al., 2014). Except for OA and BC, the
correlations for other aerosol components such as $NH_4^+$, $NO_3^-$, and $SO_4^{2-}$ were poor
(0.23, 0.13, and −0.14, respectively). The poor correlation for $SO_4^{2-}$ may result from
the large uncertainty in the emission of $SO_2$.

Because the meteorological parameters were simulated well, the simulation

discrepancies for chemical species are either caused by the emission estimation
uncertainties or by inaccuracies in the simulation of chemical oxidation processes
during LRT. Because CO, OA, and BC are accurate indicators of simulated BB
transport from Indochina (Carmical et al., 2003), the airborne measurements on board
the HALO are used as reference to evaluate the performance of the model for the flight





F0319 (Fig. 9). The 5-min merged simulation of CO concentration with (blue line) and
without (green line) BB was compared to that measured on board the HALO (red line);
the concentration was mostly in the range of 100–200 ppb, with its peak approaching
300 ppb (at 01:50, 02:50, and 04:00 UTC) at the BPTL (Fig. 9a). In general, the
simulation captured the CO variation along the flight path. However, it overestimated
the observations by nearly 100 ppb for the simulation with BB at the BPTL during
01:00–02:00, 03:40–04:20, 05:00–05:40, and 06:30–07:20 UTC (Fig. 9a). Notably, the
simulation difference was minor when the flight was in the lower part of the boundary
layer (02:30 and 06:00 UTC) i.e. < 1000m or at elevations of >4000 m (03:00–03:30
and 04:20–05:00 UTC). The model underestimated CO concentration in the lower part
of the boundary (<1000 m) (02:30 and 05:50–06:30 UTC) over the ECS. In conclusion,
our model simulation overestimates BB emissions but underestimates continental CO
emissions from China due to the underestimation of the emission inventory of the
MICS-Asia III (Kong et al.,2020) was adopted in this study.

OA and BC are also important BB indicators and were reasonably captured by the

model before 03:00 UTC when the flight was south of 28 °N at elevations of <4000 m
(Fig. 9 b-c). The time series of simulated OA and BC has peak concentrations of nearly
4-5.5 µg m$^{-3}$ and 2 µg m$^{-3}$, respectively, during HALO shuttle flights passing through
the BPTL (2000–4000 m) around 01:50 and 02:50 UTC. When BB emission was not
included in the simulation, the concentration peaks were not observed (see Fig. 9b-c,
green plot). Similar to the simulated CO results, the simulated OA and BC overestimate
the amounts of these species to the north of 30 °N at 04:00 UTC (Fig. 6a and 9). The
model after 07:30 UTC, which was related to local emission before HALO landed over
western Taiwan on 19 March 2018. In general, our model simulation captured
reasonably well OA and BC with an R of 0.55 and 0.68, respectively. A minor mean
bias for OA (BC) is 0.4 µg m$^{-3}$ (0.2 µg m$^{-3}$) and the root mean square error (RMSE) of





OA (BC) is 1.2 µg m$^{-3}$ (0.4 µg m$^{-3}$) (Table 1). The R for OA (BC) reached 0.69 (0.71),
with an RMSE of 1.0 µg m$^{-3}$ (0.5 µg m$^{-3}$) when we calculated the BB transport layer
only between 2000 and 4000 m (Table 2 and Fig. 8). In addition to OA and BC,
simulated aerosol species such as $SO_4^{2-}$, and $NH_4^+$ were overestimated, whereas $NO_3^-$
was underestimated although their concentrations were low (Table 2). Because the
BPTL was mainly between altitudes of 2000 and 4000 m, the subsequent discussion
focuses on the influence of the BPTL from Indochina on the downstream areas,
particularly the ECS and Taiwan.

**434     4.2 Effects of LRT BB plume from Indochina on East Asia**

To investigate the regional impacts of BB plume transport from Indochina, we
compared the simulation with and without BB emission for the events on 17 and 19
March 2018. The analysis of the calculations focused on the impact over SC, Taiwan
and ECS. These three selected regions are SCA (in South China), TWA (covered the
whole Taiwan), and ECSA (in the ECS) as shown in Figure 1a. After being emitted the
BB pollutants from Indochina were then transported northward to China and
subsequently northeastward. The exact flow pattern depended on the weather
conditions and flow types (ridge or trough) at 700 hPa (3000 m) between 17 and 19
March 2018 (see Fig. 2). Consequently, we investigated the hourly variation in the area
mean concentrations or mixing ratios of air pollutant trace constituents to assess the
importance of BB emissions from Indochina on the selected downstream region e.g. the
ECSA (Fig. 10), SCA, TWA and ECSA (Table 3). The contribution of CO (or others
species) due to BB was estimated by detraining the difference between simulations with
and without the BB emission. These differences are then expressed as a fraction in
percentage shown in Figure 10 (blue line). The mean concentration of CO (red line)
over the ECSA (Fig. 10a) was at its lowest (115 ppb) on 17 March 2018; it gradually
increased to a peak concentration of 280 ppb on 18 March 2018 and then remained



stable at 260 ppb on 19 March 2018. The contribution of CO from BB (blue line) ranged
from 19 % (<22 ppb) on 17 March 2018 to a peak of 42 % (~113 ppb) on 18 March
2018 and then gradually declined to 26 % on 19 March 2018 (Fig. 10a).    As for OA
(BC), the lowest percent contribution by BB was 14-16% (<5%) between 16 and 17
March 2018 while the highest could be more than 30% (60%) during 18 and 19 March
2018 (Fig. 10b and c). The variation trend of $PM_{2.5}$ its lowest percent contribution by
BB was 21 % (0.42 µg m$^{-3}$) on 17 March 2018 (Fig. 10d), increasing to 40 % (5.6 µg
m$^{-3}$) on 18 March 2018 because the BB plume spread by the strong wind to the ECSA.

The variation of $O_3$ (Fig. 10e) depends on transport and photochemistry, which

involves the precursors $NO_x$ and VOC and the photolysis frequency of $NO_2$, $J(NO_2)$.
For the elevations between 2000–4000 m, $O_3$ changes are similar to those of CO, NOx
and KET, which were mainly contributed by the LRT BB plume and related to the
ozone precursor after 18 March 2018. The lowest and highest $O_3$ concentrations on 17
and 18 March 2018 were 56 and 75 ppb, respectively, of which we estimate that 5.6
ppb (10 %) and 34 ppb (45 %) were BB's contributions, respectively. Although the
mean $NO_x$ concentration was relatively small (0.06–0.18 ppb), the BB contributed 35–
70 % (0.02–0.13 ppb) during 17–19 March 2018 (Supplementary Fig. S3a). The KET
concentration was in the range 0.4 to 2.7 ppb, with BB contributing nearly 20–26 %
(0.08–0.7 ppb) during 17–19 March 2018 (Supplementary Fig. S3b).

The area-mean OH contributed by BB increased from its lowest level (<30 %) on

17 March 2018 to its highest (nearly 70 %) on 19 March 2018 (Fig. 10f). $HO_2$ was also
observed to increase trend from 10 % to 40 % during daytime over the period 17–19
March 2018 (Supplementary Fig. S3c). The amounts of the oxidizing agent, OH, and
the free radical $HO_2$ depend on the amounts of trace gases, which produce and remove
these radicals, (eg. $NO_x$, water vapor, ozone, hydrocarbons, etc.) and the relevant
photolysis frequencies $J(O_3 \rightarrow O1D)$, $J(NO_2)$ etc.. Thus trace constituents from BB were





expected to increase OH and HO₂. However, BB's contribution to photolysis
frequencies   J(O₃→ O¹D) (Fig. 10g), J(NO₂) (Supplementary Fig. S3d) etc. decreased
as the mean BB aerosol concentration increased over the ECS during 17–19 March
2018. This is because photolysis calculation results used simulated aerosol and cloud
formation, which increased over the ECSA (Fig. 12).
The NO$_y$, mean concentration ranged from 1.0 to 4.5 ppb, of which BB's
contribution was from 55 to 82 % (Supplementary Fig. S3e). Such a high contribution
from BB also demonstrated the effects of long-distance transport. Figure 10h indicates
an increasing trend of HCHO concentration from 17 to 19 March 2018. HCHO
formation and destruction depend on the rate of reaction of OH with HCHO precursors
and the rate of reaction of HCHO with OH and the photolysis frequency of HCHO. As
a result, HCHO production varied with OH concentration. The lowest and highest
concentrations of HCHO were on 17 and 19 March 2018, respectively.   In summary,
the consistent variations in BB contributions to CO, PM$_{2.5}$, VOC, OH, NO$_x$, NO$_y$, and
O₃ peaked on 18 or 19 March 2018, whereas J(O¹D) decreased between 17 and 19
March 2018.
Figure 11 displays the fraction in % that the long-range transported BB emission
contributes to the amounts of NO$_x$, NO$_y$, PM$_{2.5}$, OA, BC, OH, O₃, CO, KET, HO₂,
HCHO and J(O¹D), over the ECSA on 17 and 19 March 2018. Except for NO$_y$, BB
contribution was generally <11 % at elevations of <1000 m over the ECSA. The scatter
distribution of the simulation results indicates that the effect of BB emission at
elevations of <1000 m (Fig. 11a) was significantly lower than that between the
elevations of 2000 and 4000 m (Fig. 11b). For NO$_y$, NO$_x$, PM$_{2.5}$, BC, OH, O₃, and CO,
the BB contribution was >30 % at the elevation of 2000–4000 m over the ECSA (Fig.
11b). Table 3 further summarizes the effect of BB emission on the downwind areas
(SCA, TWA, and the ECSA) at the <1000 m and 2000–4000 m elevations. The





contribution of BB to $NO_y$, $NO_x$, $PM_{2.5}$, BC, OH, $O_3$ and CO was at least 30–80 % at
the elevation of 2000–4000 m over the regions SCA, TWA and ECSA. In the lower
boundary layer (i.e. <1000 m), the BB contribution for most species at the remote
downstream areas was <20 %, except for TWA. Because of the high mountains (Lin et
al. 2021) present in TWA, the BB plume passing over Taiwan was potentially
transported downward through mountain–valley circulation to the lower boundary layer
(Ooi et al., 2021). The influence of BB over TWA was the highest among these three
downstream regions (see Table 3) as its location was directly on the transport pathway
for the BB plume on the major event day (flight F0319).

Figure 12 displays the simulated cloud water difference with and without BB

emission over different regions on 17 and 19 March 2018. BB aerosols are a potential
source of cloud nuclei. The simulations show the impact of BB on cloud water
enhancement (Fig. 12). Cloud water enhancement over SCA was associated with
aerosol enhancement from the BB in the altitude range 1000–4000 m: the peak being
2-2.5 mg $kg^{-1}$ at 2000 m on these 2 days (Fig. 12). The abundance of BB emissions
transported from Indochina to SCA (Fig. 3) is expected to contribute to the high cloud
water formation over SCA. Furthermore, the southerly flow (Fig. 3) that transports
warm and moist air mass from the South China Sea may have favored cloud formation
in flights F0317 and F0319. High cloud water related to BB can be seen in the
simulations of these two days. In the remote ECSA regions, the cloud water
substantially increased on 19 March 2018 (Fig. 12) compared to 17 March 2018
because of a significant difference in BB emissions transported to the ECSA between
17 and 19 March 2018 (Fig. 3). Similarly, the cloud water enhancement over Taiwan
also only appeared on 19 March 2018 (Fig. 12). Furthermore, nearly no difference in
the cloud water vertical distribution over the region IDCA (Fig. 1a) in Indochina was
noted because in the Indochina region, spring is the dry season (Lin et al., 2009) and





thus unfavorable for cloud water formation. The simulated downward short wave flux
at ground surface was 1-3% reduction over the regions ECSA and SCA (supplementary
Fig. S4) during 18-19 March 2018. The combination of BB aerosols enhancement and
increased cloud water results in shortwave radiation reduction, implying the possibility
of regional climate change in East Asia driven by BB aerosols.

**5. Summary**

The BB during spring in Indochina has a significant impact on the chemistry and

composition of the troposphere in the surrounding regions of East Asia. During the
EMeRGe campaign in Asia, atmospheric pollutants were measured on board the HALO
aircraft. In this study, a minor long-range BB transport event was observed from
Indochina on 17 March 2018 (flight F0317), followed by a major long-range BB
transport event on 19 March 2018 (flight F0319). The impact on tropospheric trace
constituent composition and the environment has been investigated.

During the major BB transport event F0319, the 1-min mean of the peak

concentrations of the trace constituents CO, O3, ACE, ACN, OA and BC between the
altitudes of 2000 and 4000 m over the ECS were 312.0 ppb, 79.0 ppb, 3.0 ppb, 0.6 ppb,
6.4 $\mu$g m$^{-3}$, 2.5 $\mu$g m$^{-3}$ respectively. In comparison during the F0317 event CO, O3,
ACE, ACN, OA and BC were 203.0 ppb, 71.0 ppb, 2.0 ppb, 0.3 ppb, 3.4 $\mu$g m$^{-3}$, 1.2
$\mu$g m$^{-3}$ respectively.

When the elevation was <1000 m for both the F0317 and F0319 events, the sulfates,

rather than OA, had the highest concentrations. The peak concentration could be as high
as 5.1 $\mu$g m$^{-3}$ in the low boundary for the event F0317 in the ECS. This observation is
most likely explained by a continental outflow from regions having fossil fuel
combustion in the lower boundary layer over the ECS.

In this study, the WRF-Chem model was employed to evaluate the BB plume

transported from Indochina and its influence on the downstream areas including South
China, Taiwan, and the ECS. The contribution of the BB plume for most species in the
remote downstream areas was <20 % in the lower boundary layer (altitude <1000 m).
In comparison, the contribution of long-range transported BB plume was 30–80 %, or
even higher, for many of the trace constituents ($NO_y$, $NO_x$, CO, OH, $O_3$, BC and $PM_{2.5}$)
in the altitude range between 2000 and 4000 m for SC, Taiwan, and the ECS. The large
influence of BB over Taiwan is most probably because the BB transport passes directly
over Taiwan.
BB aerosols are potential sources of cloud nuclei. The WRF simulations estimate
the effect of the BB plume on cloud water formation over SC and the ECS. We observe
in the simulations cloud water enhancement over SC at elevations of 1000–4000 m.
This increase of cloud water is consistent with an increase in aerosol, caused by BB
emissions, transported from Indochina to SC. In remote regions of the ECS, the
simulated cloud water was significantly larger during the major BB event on 19 March
2018 than the minor BB event on 17 March 2018. The simulated decrease of the
photolysis frequency ($J(O^1D)$ and $J (NO_2)$) is attributed to the difference in aerosol
concentrations and associated cloud enhancement between the two events over the ECS.
This we explain by the significant differences in BB emissions transported to the ECS
between the two events.
Interestingly, we found the combination of increased BB aerosol concentration
and increased amounts of cloud water led to reductions in the amount of incoming
shortwave radiation at the surface over the ECS and SC. This influences tropospheric
chemistry and composition, regional climate, precipitation, ocean biogeochemistry,
agriculture and human health.

***Data availability***



The EMeRGe data are available at the HALO database
(https://doi.org/10.17616/R39Q0T, DLR, 2022) and can be accessed upon registration.
Modeling data can be made available upon request to the corresponding author.
*Author contribution*
CYL conceived the idea, analyzed the data, writing and editing of the manuscript. WNC
and YYC run the model and analyzed the data. CKC joined the manuscript
discussion.CYLiu provided the MODIS data. HZ and HS provided trace gases data. EF
provided acetonitrile data. FO performed the ozone measurement. OOK, BAH and
MLP were responsible for the BC measurement. KK and JS were responsible for C-
ToF-MS measurements. JPB and MDAH led the EMeRGe-Asia experiment. All
authors have read and agree to the published version of the manuscript.
*Competing interests*
The authors declare that they have no conflict of interest.
*Acknowledgments:*
The accomplishment of this work has financial support from the Ministry of Science
and Technology, Taiwan, under grants MOST 108-2111-M-001-002, 109-2111-M-001-
004 and 110-2111-M-001-013. We thank to National Center for High-performance
Computing (NCHC) for providing computational and storage resources.
The HALO deployment during EMeRGe was funded by a consortium comprising the
German Research Foundation (DFG) Priority Program HALO-SPP 1294, the Institute
of Atmospheric Physics of DLR, the Max Planck Society (MPG), and the Helmholtz
Association. Johannes Schneider and Katharina Kaiser acknowledge funding through
the DFG (project no. 316589531).

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










Table 1 Observed and simulated mean values for bias (BIAS), root mean square error
(RMSE), and correlation coefficients (R) for EMeRGe HALO flights on 17 and 19
March 2018. KET*: the observed Acetone is applied to compare with simulated ketones
(KET).

| | OBS_ave | SIM_ave | BIAS | RMSE | R |
|---|---|---|---|---|---|
| THETA(K) | 304.8 | 304.2 | -0.6 | 1.1 | 0.99 |
| WS(m/s) | 9.1 | 8.5 | -0.6 | 2.0 | 0.94 |
| RH(%) | 63.6 | 63.0 | -0.6 | 10.6 | 0.92 |
| OA($\mu$g/m$^3$) | 1.2 | 1.5 | 0.4 | 1.2 | 0.55 |
| BC($\mu$g/m$^3$) | 0.4 | 0.5 | 0.2 | 0.4 | 0.68 |
| SO$_4^{2-}$($\mu$g/m$^3$) | 1.1 | 2.5 | 1.4 | 2.2 | 0.37 |
| NO$_3^-$($\mu$g/m$^3$) | 0.2 | 0.6 | 0.5 | 2.1 | 0.31 |
| NH$_4^+$($\mu$g/m$^3$) | 0.4 | 0.7 | 0.3 | 1.1 | 0.49 |
| CO(ppb) | 170.8 | 188.9 | 18.1 | 69.6 | 0.46 |
| SO$_2$(ppb) | 0.2 | 0.9 | 0.7 | 1.3 | 0.52 |
| O$_3$(ppb) | 59.7 | 63.2 | 3.5 | 14.1 | 0.42 |
| NO$_x$(ppb) | 0.2 | 0.2 | 0.0 | 0.2 | 0.74 |
| NO$_y$(ppb) | 1.2 | 2.8 | 1.5 | 2.5 | 0.04 |
| KET*(ppb) | 1.4 | 1.5 | 0.1 | 0.9 | 0.59 |
| TOL(ppb) | 0.1 | 0.1 | 0.0 | 0.1 | 0.76 |
| XYL(ppb) | 0.1 | 0.0 | 0.0 | 0.1 | 0.38 |
| HCHO(ppb) | 0.1 | 0.7 | 0.6 | 0.7 | 0.50 |
| HONO(ppt) | 10.5 | 1.0 | -9.4 | 15.4 | 0.55 |






Table 2 Observed and simulated mean values at an elevation between 2 km and 4 km
for bias (BIAS), root mean square error (RMSE), and correlation coefficients (R) during
EMeRGe HALO flights on 17 and 19 March 2018. KET*: the observed Acetone is
applied to compare with simulated ketones (KET).

|  | OBS_ave | SIM_ave | BIAS | RMSE | R |
|---|---|---|---|---|---|
| THETA(K) | 307.5 | 306.7 | -0.8 | 0.9 | 0.98 |
| WS(m/s) | 8.2 | 7.9 | -0.3 | 1.7 | 0.93 |
| RH(%) | 55.8 | 56.1 | 0.4 | 7.6 | 0.96 |
| OA($\mu$g/m$^3$) | 1.3 | 1.8 | 0.5 | 1.0 | 0.69 |
| BC($\mu$g/m$^3$) | 0.4 | 0.7 | 0.3 | 0.5 | 0.71 |
| SO$_4^{2-}$($\mu$g/m$^3$) | 0.8 | 2.6 | 1.8 | 2.3 | -0.14 |
| NO$_3^-$($\mu$g/m$^3$) | 0.1 | 0.1 | -0.1 | 0.4 | 0.13 |
| NH$_4^+$($\mu$g/m$^3$) | 0.4 | 0.4 | 0.1 | 0.3 | 0.23 |
| CO(ppb) | 164.4 | 223.4 | 59.0 | 80.3 | 0.60 |
| SO$_2$(ppb) | 0.0 | 1.0 | 1.0 | 1.2 | -0.03 |
| O$_3$(ppb) | 60.1 | 72.2 | 12.1 | 14.5 | 0.54 |
| NO$_x$(ppb) | 0.1 | 0.2 | 0.0 | 0.1 | 0.54 |
| NO$_y$(ppb) | 1.0 | 3.8 | 2.8 | 3.3 | 0.53 |
| KET$^*$(ppb) | 1.5 | 1.9 | 0.4 | 0.9 | 0.71 |
| TOL(ppb) | 0.1 | 0.0 | 0.0 | 0.1 | 0.13 |
| XYL(ppb) | 0.0 | 0.0 | 0.0 | 0.0 | -0.17 |
| HCHO(ppb) | 0.1 | 0.7 | 0.6 | 0.8 | 0.23 |
| HONO(ppt) | 6.0 | 0.6 | -5.4 | 7.2 | 0.24 |






Table 3: Simulated biomass burning contribution (with and without BB emission in
Indochina) in percentage (%) on 17 and 19 March, 2018 for different regions: SCA,
TWA, ECSA as shown in Figure 1a

| | SCA | | TWA | | ECSA | |
|---|---|---|---|---|---|---|
| Average | < 1KM | 2-4KM | < 1KM | 2-4KM | < 1KM | 2-4KM |
| $NO_y$ | 14.9 | 72.9 | 44.8 | 83.2 | 18.9 | 71.5 |
| $NO_x$ | -1.4 | 58.1 | 3.7 | 70.8 | 2.1 | 51.2 |
| $PM_{25}$ | 5.8 | 44.6 | 14.7 | 56.4 | 7.2 | 31.9 |
| OA | 4.4 | 37.2 | 6.7 | 47.6 | 4.6 | 24.6 |
| BC | 6.9 | 74.4 | 14.0 | 81.0 | 6.1 | 42.4 |
| OH | 14.3 | 43.6 | 24.7 | 66.6 | 10.0 | 48.1 |
| O3 | 18.8 | 33.7 | 23.5 | 38.5 | 9.4 | 31.0 |
| CO | 9.8 | 31.4 | 21.7 | 37.8 | 11.1 | 32.0 |
| KET | 6.0 | 17.0 | 9.0 | 26.8 | 7.0 | 24.5 |
| HCHO | -3.5 | 10.2 | -3.8 | 20.7 | -4.0 | 10.3 |
| $HO_2$ | 9.0 | 4.5 | 15.4 | 35.5 | 6.4 | 24.9 |
| $J(O^1D)$ | -3.0 | -1.7 | -1.1 | 0.4 | -1.6 | -1.1 |


(a)
(b)

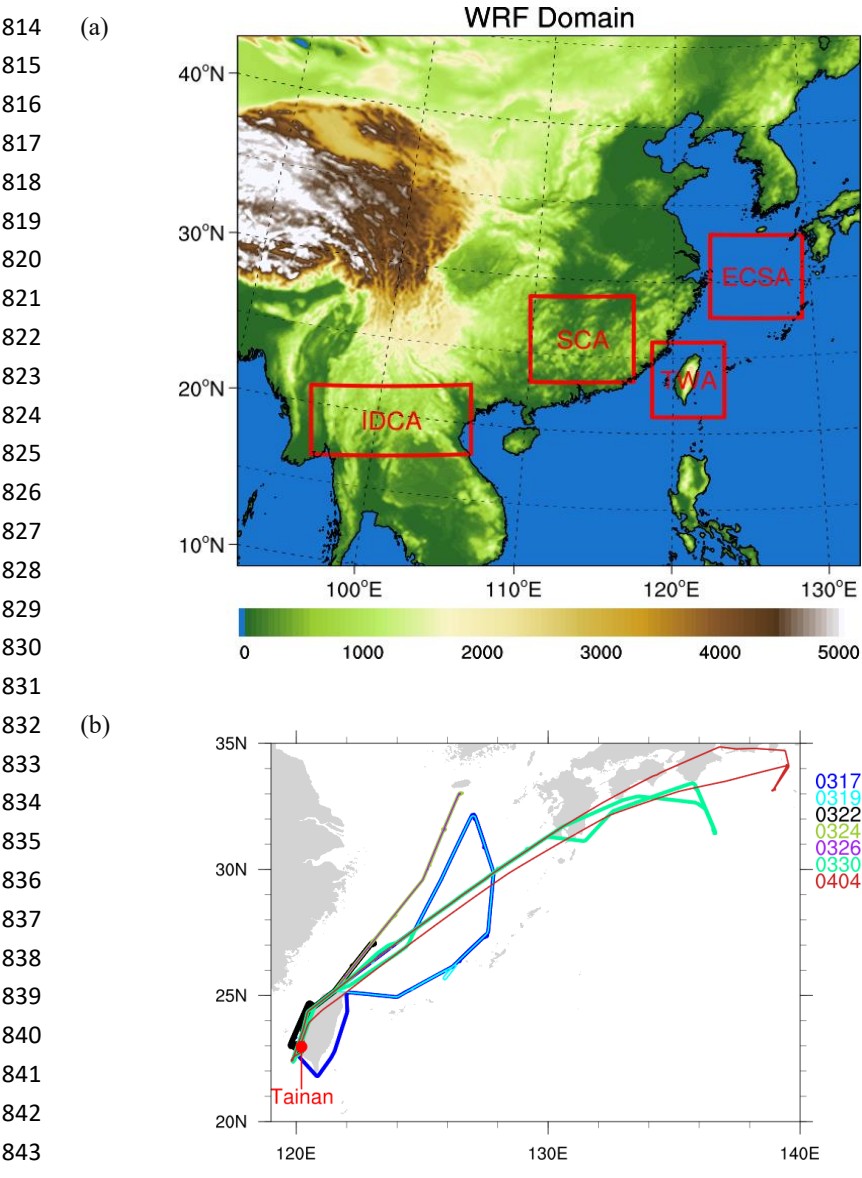

Figure 1 (a)   Configuration of Weather Research and Forecasting model domain,
topography, and location of proposed study areas in East Asia, namely IDCA (Indochina
area), SCA (southern China area), TWA (Taiwan area) and ECSA (East China Sea area,
respectively.   (b) The HALO flights on 17, 19, 22, 24, 26, 30 March, and 04 April
during EMeRGe Asia campaign. Different colors indicated different flights over East
Asia. Maps and plots were produced using NCAR Command Language (NCL) version
850 6.6.2.

(a)


(c)

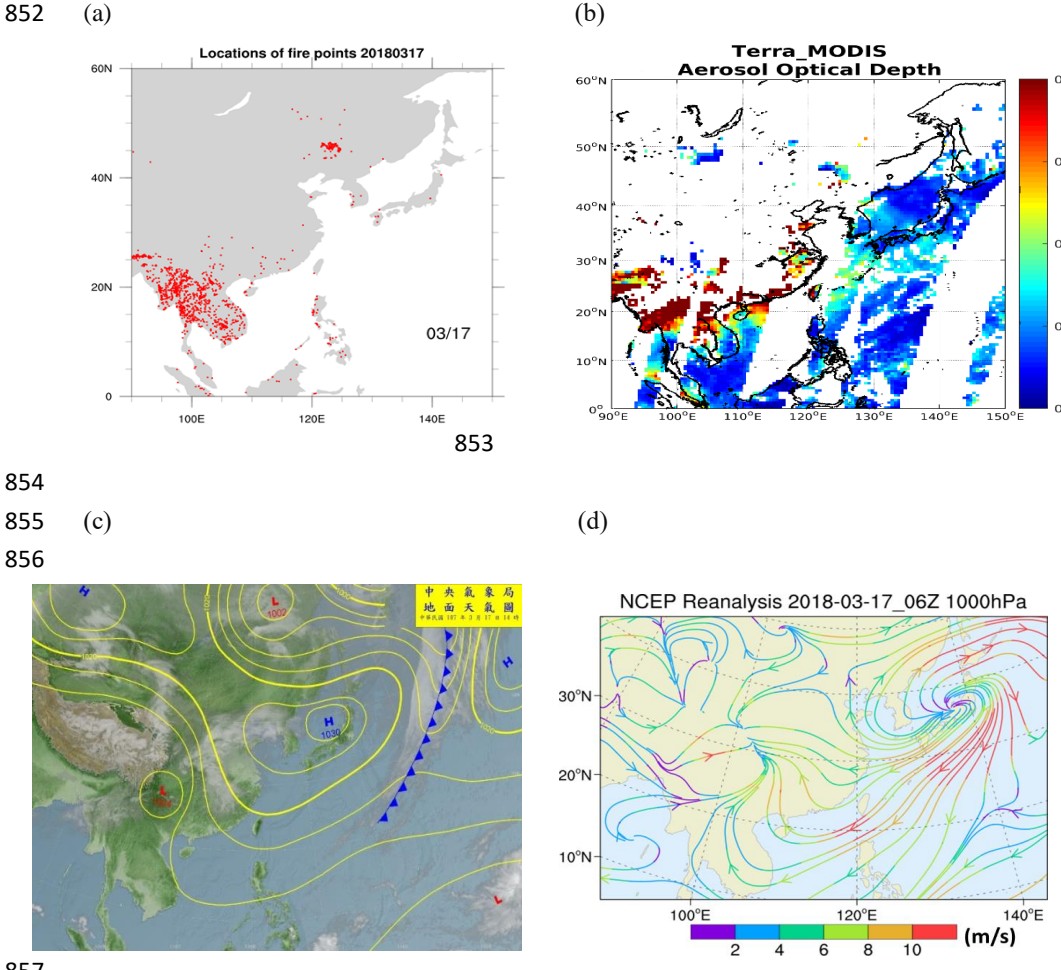


Fig.2 (a) MODIS fire hot spots on 17 March 2018 (source: https://modis-
fire.umd.edu/guides.html) and (b) Composited Aerosol Optical Depth (AOD) from
MODIS onboard NASA Terra satellite. The Collection 6.1 AOD is downloaded from
NASA Earth Data website (https://www.earthdata.nasa.gov/learn/find-data), and
composted for 0110, 0115, 0120, 0125, 0130, 0250, 0255, 0300, 0305, 0310, 0430,
0435, 0440, 0445, 0610, 0615, 0620, 0745 and 0750UTC data granules on 17 March
2018. (c) weather Chart at 06:00 UTC on 17 March 2018 (d) 1000 hPa streamlines at
06:00 UTC, 17 March 2018 (e) and (f) same as (c) and (d) but on 19 March 2018 ;(g)
700 hPa streamlines at 06:00 UTC, on 17 March 2018 (h) 700 hPa geopotential height
at 06:00 UTC, on 17 March 2018;   (i) and (j) same as (g) and (h) but on 19 March
868 2018.

Near-surface weather charts and satellite images were provided by Central Weather
Bureau (CWB) Taiwan. The near-surface and 700 hPa streamlines and geopotential



height were deduced from NCEP Reanalysis data. Maps and plots were produced using
NCAR Command Language (NCL) version 6.6.2.

(e)                                                          (f)

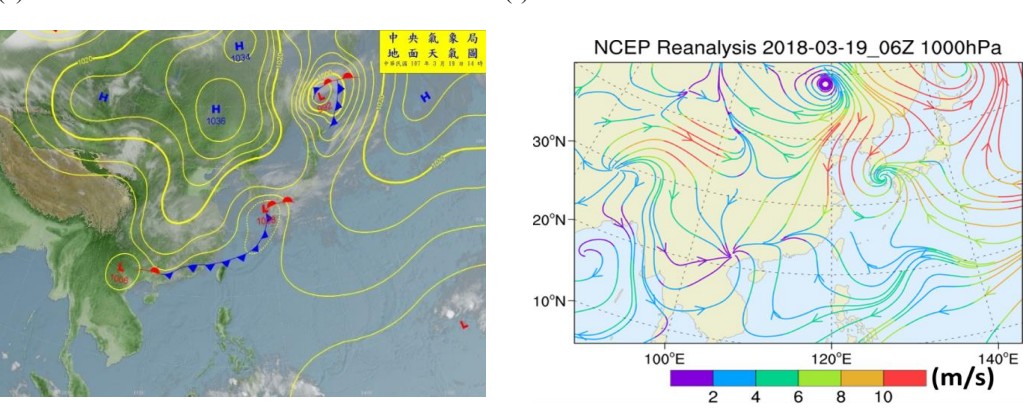



(g)                                                          (h)

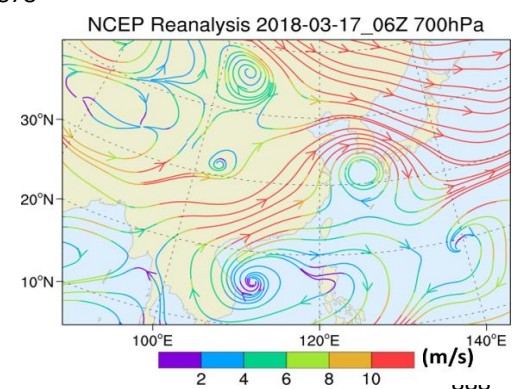

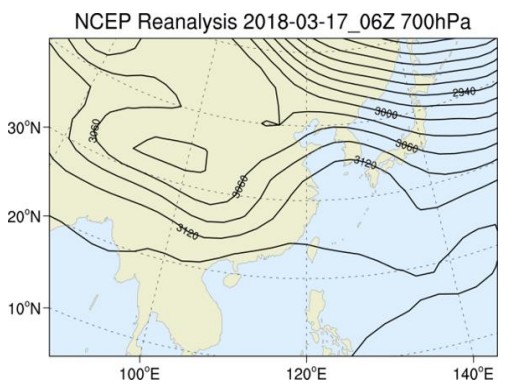




Figure 2 e-h continued





(i)                                                          (j)






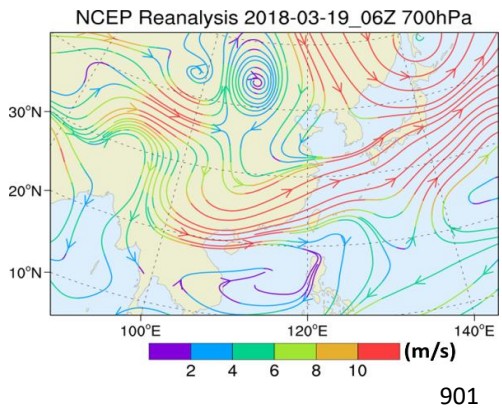

Fig. 2 i-j continued

927 (a)        (b)

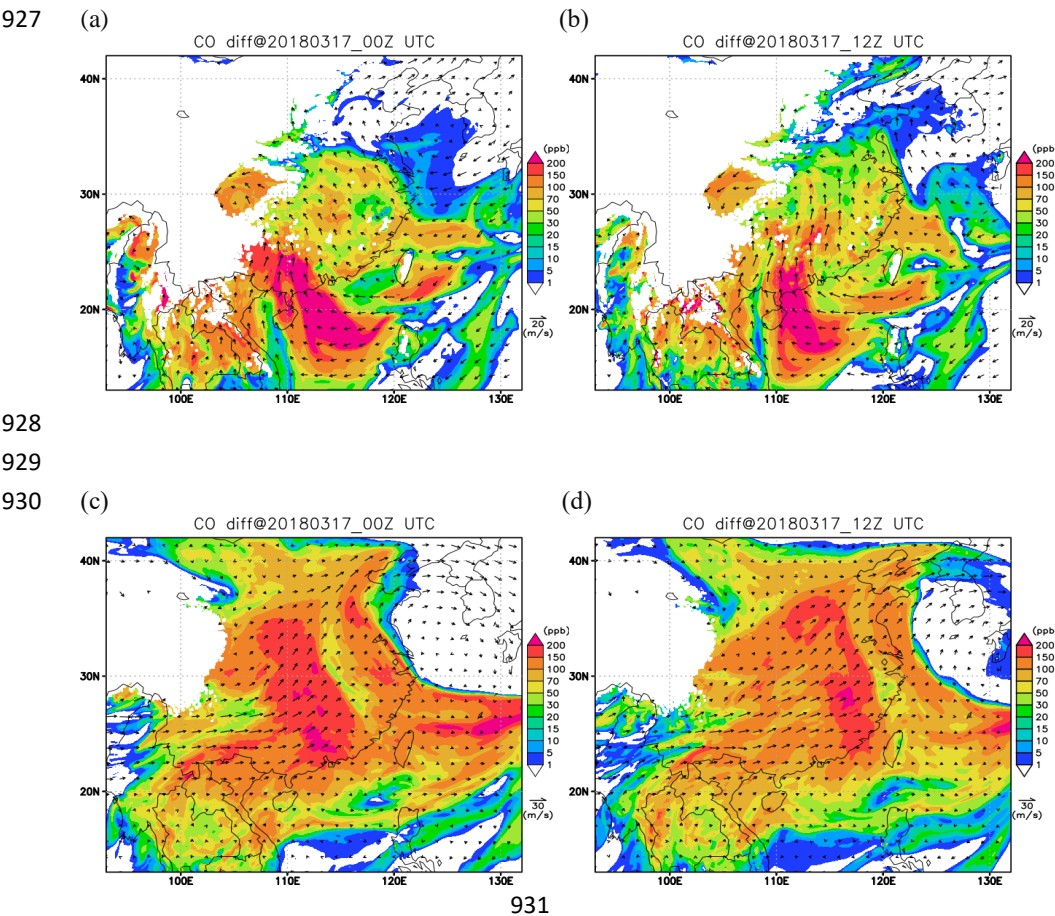



930 (c)        (d)




934 Fig 3 a-d : Simulated wind field (m s⁻¹) distribution and concentration (unit: ppb)

935 difference with and without BB emission for CO on 17 March, 2018 at 00:00 UTC (a,

936 c) and 12:00 UTC (b, d) for 1km altitude (a, b) and 3km altitude (c, d). (unit:ppb)





(e)                                     (f)

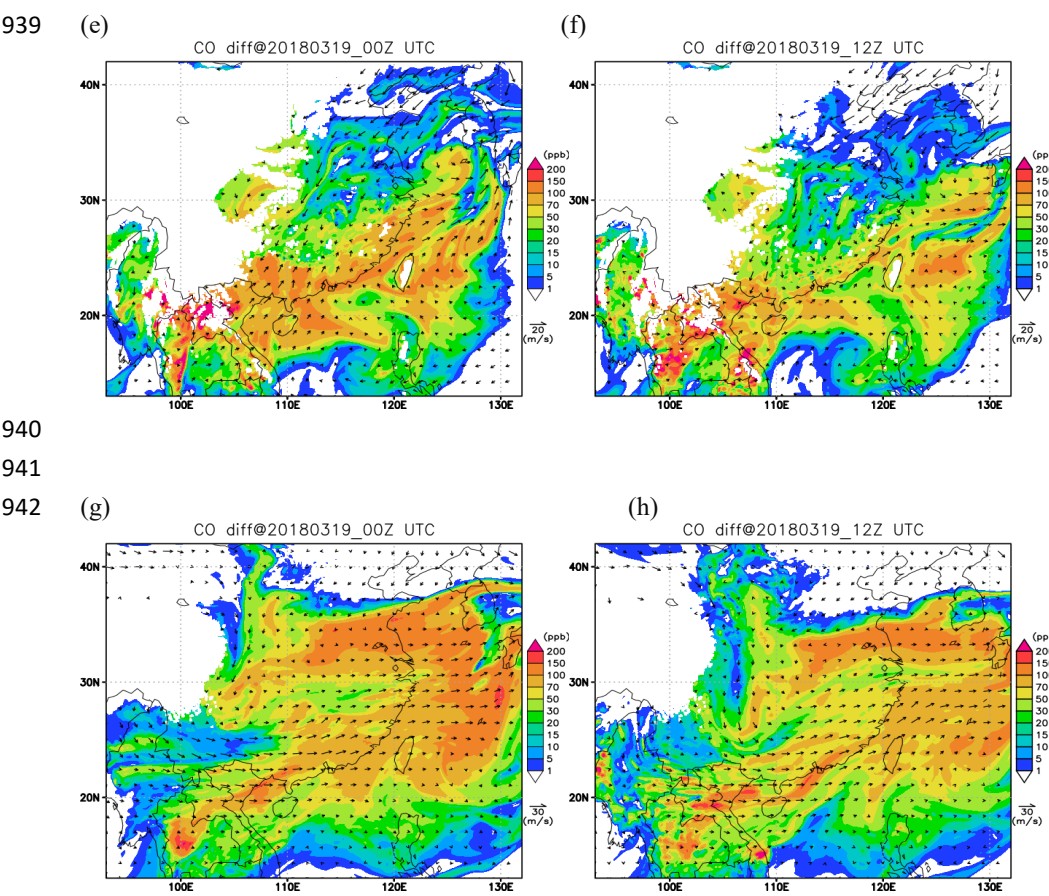



(g)                                     (h)

Fig 3 e-h: Simulated wind field (m s$^{-1}$) and concentration (unit: ppb) difference with
and without BB emission for CO on 19 March, 2018 at 00:00 UTC (e, g) and 12:00
UTC (f, h) for 1km altitude (e, f) and 3km altitude (g, h).











(a)






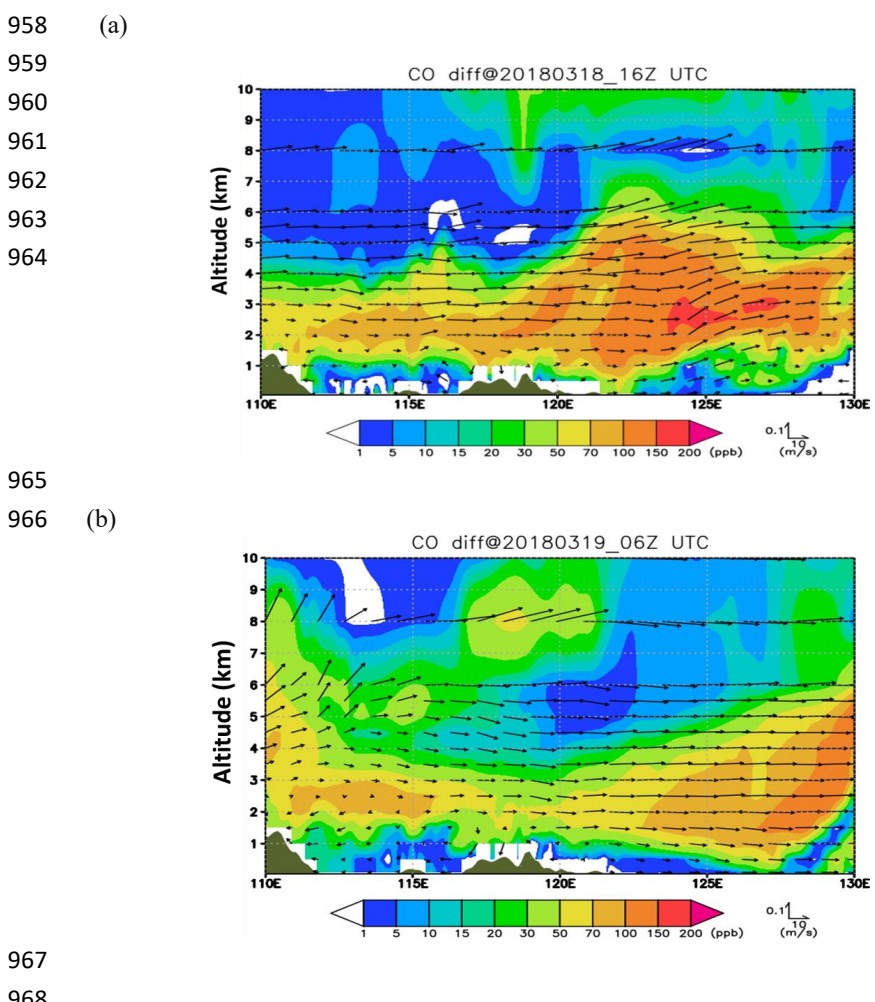


(b)




Fig. 4 Simulated wind field (m s$^{-1}$) distribution and the concentration (ppb) difference
between with and without BB emission for CO at cross-section 30 °N (a) 16:00 UTC
18 March 2018 (b) 06:00 UTC, 19 March 2018. Wind vectors represent along section
winds, with scales shown at the down-right corner of plot (unit: m s$^{-1}$)







Fig. 5 (a) The HALO flight and detailed locations on 17 March 2018. (b) Flight altitude and 1-min mean of observed concentrations for CO (upper), Organic aerosol (OA), BC aerosol (BC), $SO_4^{2-}$, $NO_3^-$, $NH_4^+$ (middle), $O_3$, acetone (ACE) and acetonitrile (ACN) (bottom) on 17 March.   (c) The observed $SO_4^{2-}$ mass concentration by HALO along with height-latitude variations on 17 March 2018 (d) The observed OA mass concentration by HALO along with height-latitude variations on 17 March 2018 (e) Result of the HYSPLIT model backward trajectory analysis started at the location of the HALO flight path at 02:00, 04:00, 06:00, 09:00 UTC on 17 March 2018.

Figure 5 c-e

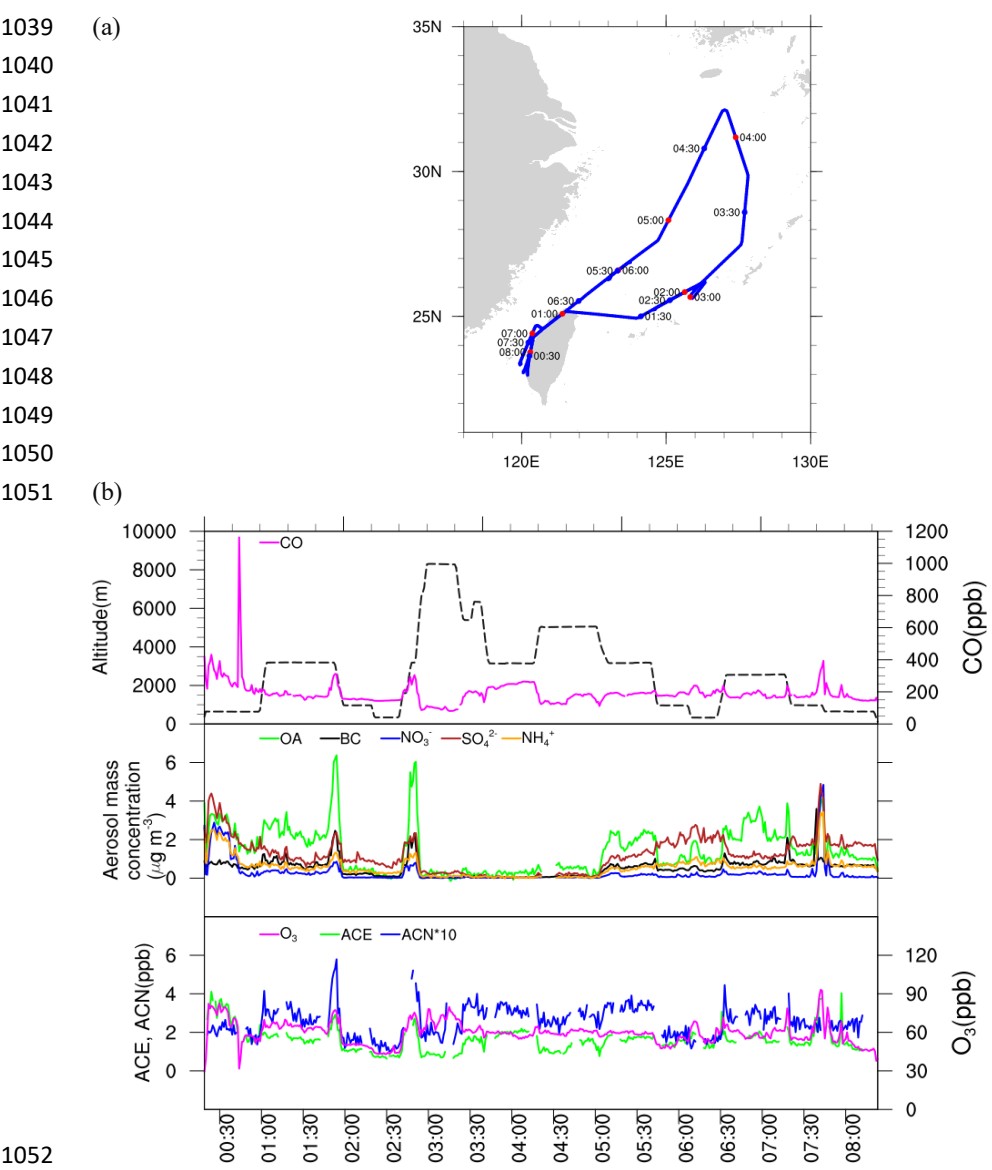

(a)
(b)
Figure 6 (a) The HALO flight and detailed locations on 19 March. (b) Flight altitude
and 1-min mean of observed concentrations for CO (upper), Organic aerosol (OA), BC
aerosol (BC), $SO_4^{2-}$, $NO_3^-$, $NH_4^+$ (middle), O3, acetone (ACE) and Acetonitrile (ACN)
(bottom) on 19 March 2018.   (c) The observed $SO_4^{2-}$ mass concentration by HALO
along with height-latitude variations on 19 March 2018 (d) The observed OA mass
concentration by HALO along with height-latitude variations on 19 March 2018 (e)
Result of the HYSPLIT model backward trajectory analysis started at the location of
the HALO flight path at 02:00, 04:00, 05:00, 07:00 UTC on 19 March 2018.





(c)
(d)
(e)
Figure 6 c-e



(a) (b)

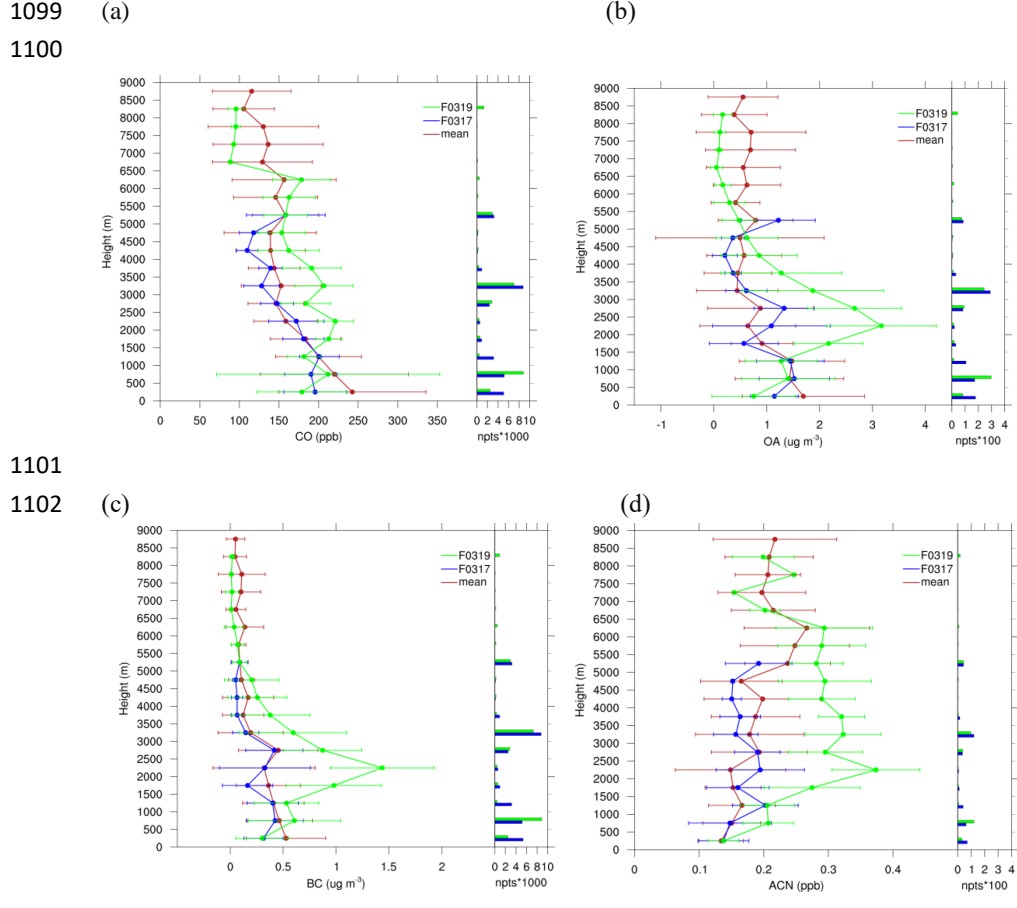

(c) (d)

Fig.7 Observed vertical distribution calculated as 1-min mean and 500 m interval with one standard deviation of the concentrations for the mean profiles (red) (including 17, 19, 22, 24, 26, 30 March, and 04 April 2018) and flights on 17 (blue) and 19 (green) March 2018. (a) CO (b) OA (c) BC (d) Acetonitrile (ACN) (e) Acetone (ACE) (f) $O_3$ (g) J ($O^1D$) (h) $NO_y$. The number of data points is shown in the right panel.

(e)

(g)

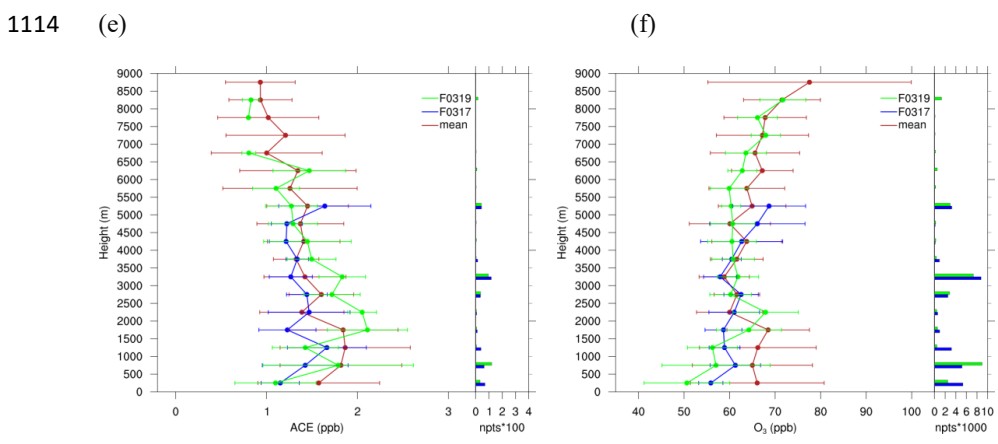







Figure 7 continued










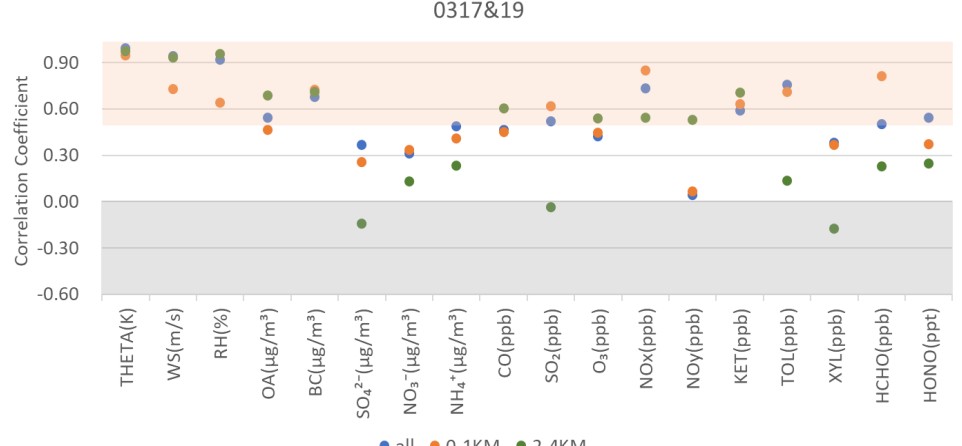



Fig. 8    Correlation Coefficient (R) between observation and simulation along with the
HALO flights at the elevations 0-1 km, 2-4 km, and the whole track (all) on 17 and 19
March 2018.




















(a)

(b)

(c)

Fig.9 Observed (OBS, red) and simulated concentration with (CTRL, blue) and without (noBB, green) BB emission along with the flight altitude for (a) CO (ppb) (b) OA ($\mu g\,m^{-3}$) (c) BC ($\mu g\,m^{-3}$) on 19 March 2018.



(a)
(b)
(c)
(d)
(e)
(f)
(g)
(h)

Figure 10 Hourly variation of simulated mean concentration (red) and contributed by
BB (%, blue) between 2 km and 4 km over the region ECSA in Fig.1a during 15-19
March 2018. (a) CO (b) OA (c) BC (d) $PM_{2.5}$ (e)$O_3$ (f) OH (g) J($O^1$D), and (h) HCHO



(a)

(b)

Figure 11 Box plots of simulated BB influences (%) on $NO_y$, $NO_x$, $PM_{2.5}$, OA, BC, OH, $O_3$, CO, KET, HCHO, $HO_2$ ,and $J(O^1 D)$ over the region ECSA in Fig. 1a on 17 and 19 March 2018. (a) below 1 km, (b) between 2 km and 4 km




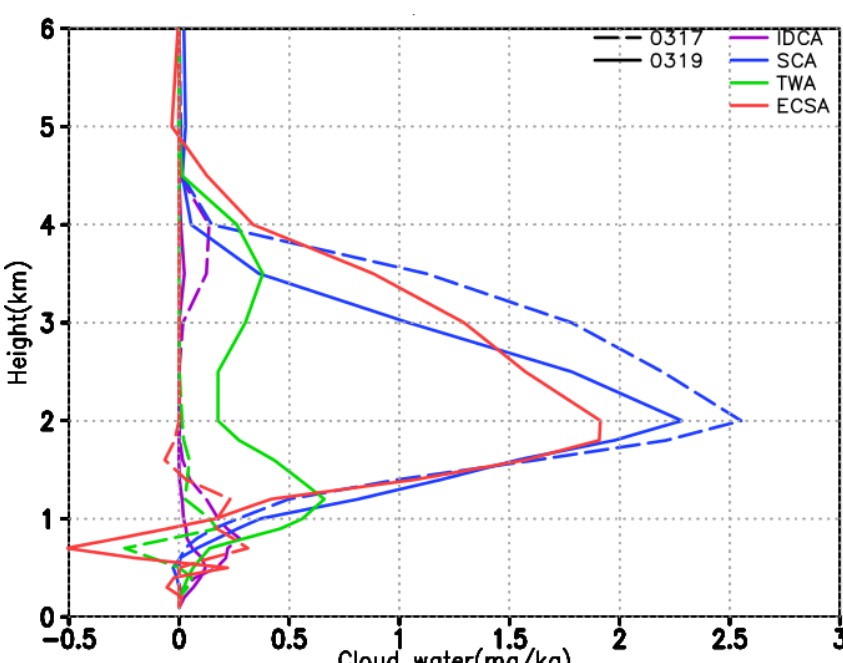

Figure 12    Simulated vertical distribution of BB influences on cloud water difference
between with and without BB emission on 17 (dash) and 19 (solid) March 2018.
Regions include IDCA, SCA, TWA, and ECSA as shown in Figure 1a.