# Peer review of "Effects of transport on a biomass burning plume from Indochina during EMeRGe-Asia identified by WRF-Chem"

_Atmospheric Chemistry and Physics, 2022_

## Author Response (AR1)

Response to Reviewer#1

1. The fire plume rise and its simulation by WRF-Chem aren't discussed in the paper, thought there is extensive analysis of the vertical distribution of the biomass burning plumes. The fire plume rise is an important process by which aerosol and gaseous species from wildland fires are injected into the free troposphere, where they can be transported to long distances. WRF-Chem has an 1D plume rise parameterization (Freitas et al.). It isn't clear if this scheme was used, and how it performed, associated uncertainties and their impact on the findings on the results presented here.

R: Thank you for your comments. Yes, the Freitas et al (2007) 1-D plume-rise model has been incorporated into WRF-Chem (Power et al. 2017; Grell et al. 2011), and this scheme was used in our simulation (Please also see Q2(a) in this response). To estimate heat flux, fires are divided into four surface categories based on WRF's land use dataset: savanna, grassland, tropical and extra-tropical forest. Simulated atmospheric sounding data for the plume rise model are computed every hour at each grid point containing an active fire. The final height reached by a plume is controlled by the thermodynamic stability of the atmospheric environment and the surface flux released from the fire (Freitas et al. 2011; Grell et al. 2011).

As mentioned in the article, our previous study (Lin et al. 2009) proposed that the mountain lee-side trough over Indochina plays a dominant role in the uplift of the BB pollution easily transport to the elevation >3000 m. Actually, this mechanism also can be applied in this case, as shown in Figure R1 (a) and (b), weak wind speed (near calm) and stable weather conditions existed in the boundary layer from the sounding (WMO 48327, Figure R1(a) and (b)) at Chiang-Mai, Thailand. According to the 850 and 700 hPa weather map (Fig. R1 (c) and (d)), a clear lee side trough formed between 16-17 March 2018 might provide an extra force for uplifting the air mass to a high elevation. Therefore, it is not easy to estimate the impact of individual factors on the plume rise height. The uncertainties of the plume injection height were not only related to the number of fire hot spots, and land use (surface categories), but also weather conditions (e.g., lee side trough existed or not).

[Figure]

Figure R1 Sounding at station 48327 (located at ChingMei, Thailand) at (a) 00:00UTC, 16 March (b) 00:00 UTC, 17 March 2018.; The weather charts from the Central Weather Burea of Taiwan at 12 UTC on 16 March 2018 (c) 850 hPa (d) 700 hPa

**References:**

Freitas, S. R., Longo, K. M., Chatfield, R., Latham, D., Silva Dias, M. A. F., Andreae, M. O., Prins, E., Santos, J. C., Gielow, R., and Carvalho Jr., J. A.: Including the sub-grid scale plume rise of vegetation fires in low resolution atmospheric transport models, Atmos. Chem. Phys., 7, 3385–3398, doi:10.5194/acp-7-3385-2007, 2007.

Freitas, S. R., Longo, K. M., Alonso, M. F., Pirre, M., Marecal, V., Grell, G., Stockler, R.,
Mello, R. F., and S´anchez G´acita, M.: PREP-CHEM-SRC – 1.0: a preprocessor of trace
gas and aerosol emission fields for regional and global atmospheric chemistry models,
Geosci. Model Dev., 4, 419–433, doi:10.5194/gmd-4-419-2011, 2011.

Grell, G., Freitas S.R., Stuefer M., Fast, J., Inclusion of biomass burning in WRF-Chem:
impact of wildfires on weather forecasts. Atmos. Chem. Phys., 11, 5289–5303,
doi:10.5194/acp-11-5289-2011, 2011.

Lin, C.Y., H.M. Hsu, Y.H. Lee, C. H. Kuo, Y.F. Sheng, D. A. Chu: A new transport
mechanism of biomass burning from Indochina as identified by modeling studies., Atmos.
Chem. Phys., 9, 7901-7911. DOI: 10.5194/acp-9-7901-2009 https://doi.org/10.5194/acp-9-
7901-2009, 2009

Powers G., Klemp J. B. , Skamarock W. C., Davis C. A., Dudhia J., Gill D. O., et al.The weather
research and forecasting model overview, System Efforts, and Future Directions.Bulletin of
the American Meteorological Society 2017 Vol. 98 Issue 8 Pages 1717-1737

2. (a) Figure 12 shows the effect of the smoke plume on cloud water. However,
   I can't find any description of the model configuration on how the aerosol
   feedback on the meteorology is simulated in this study.

   R: Thank you for the suggestions. Table R1 (also see Table 1 in the revision article)
   summarizes the model configuration in this revision. Regarding the aerosol indirect
   effect and the discussion on aerosol feedback, please see the responses in the next
   question (Q2 (b) ) and Q4.

| | |
|---|---|
| Resolution | 10km |
| Microphysics | Lin |
| Cumulus parameterization | Grell 3D ensemble scheme |
| Planetary Boundary Layer | Mellor-Yamada-Janjic TKE scheme |
| Longwave radiation | RRTMG |
| Shortwave radiation | RRTMG |
| Fire emissions | FINN V1.5 |
| Anthropogenic emissions | MICS-Asia III(2010) + Taiwan Emission Data System ver 9.0 (2013) |
| Biogenic emissions | MEGAN V2.04 |
| Chemistry option | RACM Chemistry with MADE/VBS aerosols using KPP library along with the volatility basis set (VBS) used for Secondary Organic Aerosols |
| Photolysis option | Madronich |
| wet scavenging | On , (Neu and Prather, 2012) |
| Cloud chemistry | On, |
| feedback from the aerosols to the radiation schemes | On |
| the time interval for calling the biomass-burning plume rise subroutine | 180 min |
| feedback from the parameterized convection to the atmospheric radiation and the photolysis schemes | On |
| Subgrid-scale wet scavenging | on |
| Subgrid aqueous chemistry | on |

Table R1 WRF-Chem model
configuration and physics
and chemistry options in this
study. (RRTMG=Rapid
Radiative Transfer Model for
General Circulation
Models;FINN=Fire
Inventory from National
Center for Atmospheric
Research)

(b) There are several feedback mechanisms of the aerosols on meteorology. Although WRF-Chem contains a few parameterizations to simulate these processes, large uncertainties remain with respect accurate representation of the aerosol-radiation-microphysics interactions. Authors present the results for such complex phenomena in a single graph without thorough discussion and sensitivity analysis (e.g. direct vs. indirect feedback). Moreover, given the relatively low aerosol concentrations in the smoke plumes analyzed here the sensitivity of the simulated cloud water concentrations to smoke plumes seem to be overly large.

R: Thank you for the suggestions. We agree that large uncertainties remain exited and still a challenge to accurately represent the aerosol-radiation-microphysics interactions, even in the state-of-the-art numerical model. Therefore, our intended purpose is trying to evaluate the potential impacts of long-range transport BB pollution from Indochina on East Asia.

Although the concentrations of individual aerosol components are low, the sum of measurement major aerosol components ($OA+BC+SO_4^{2-}+NO_3^-+NH_4^+$) on 19 March at BPTL (BB plume transport layer, 2000-4000 m) could be more than 13 µg/m$^3$ (Fig. 6b). Thus, the related feedback could be significant. As suggested by the reviewer, we further did the sensitivity analysis for the direct and indirect effects and related short-wave reduction at the ground surface in this revision.

Fig R2a shows the results along the flight for the simulated OA of the control simulation (CTRL, i.e. running with aerosol direct and indirect effect) and the case ROCD (Running Only Considered Direct effect). As mentioned in the article, a frontal system was just located from the East China Sea (ECS) to Southern China (SC) (Fig. 2e) on the event day, 19 March 2018. Most of the time the difference was not significant between the CTRL and the case ROCD, except for during 03:30-04:20 UTC where the flight was located north of 28 $^\circ$N (Fig. R2b) and a frontal cloud band existed (Fig. 2g). In other words, the effect of wet scavenging reduced the aerosol concentration bias in the ECS and SC, as for the frontal system providing the moist air mass in the event flight F0319. **(L436-441)** (also see Figure 9 in the revised article)

We also carefully checked the hourly variations of the impact of aerosol indirect effect in the simulation. Figures R3 (a)-(c) indicate the simulated spatial distribution of PM$_{2.5}$ and cloud water for the CTRL simulation at an altitude of 2000 m at 02:00, 04:00, and 06:00 UTC, respectively. The simulated cloud water area could be represented by the location of the frontal system, from Korea and Japan to southern China. First of all, we examine the role of the indirect effect (chemistry-microphysics interactions) in the

simulation. Figure R3 (d)-(f) showed the difference between the control simulation (case CTRL) and the case ROCD. It was noted that wet scavenging mainly occurred along the frontal system and north of it, from Japan to southern China, i.e. major impacts were over ECS and SC. These results are also consistent with the finding in Figure 12 and the results in Figure 9. The impact of the BB (i.e. difference between with (ctrl run) and without BB emission) on these regions was shown in Figure R3 (g)-(i). The results indicated that the BB plume was transported mainly south of 30 N and the enhancement area of cloud water was along the frontal system. Figure R4 shows the cloud water difference when the aerosol indirect effect turned off in the simulation over different regions on 19 March 2018. The significant cloud water shortage over ECSA, and SCA could be as high as 2.4 mg/kg and 1.5 mg/kg, respectively. In other words, the role of the chemistry-microphysics interactions (indirect effect) plays an important role in the cloud water enhancement in the SCA and ECSA in this study **(L549-554)** (also see Figure 12b in the revised article).

Figure R5 shows the simulated downward shortwave flux at the noontime at ground surface due to BB was 2-4% and 5-7% reduction over the regions ECSA and SCA, respectively, during 18-19 March 2018. However, a significant shortwave flux reduction at noontime at the ground surface could be 15-20% due to aerosol indirect effect in the region SCA during 18-19 March 2018. The combination of BB aerosols enhancement and increased cloud water results in shortwave radiation reduction, implying the possibility of regional climate change in East Asia driven by BB aerosols. **(L557-560)** (also see supplementary Fig. 4a-b in the revised article).

[Figure]

Figure R2 (a) Observed (OBS, red) and simulated concentration with (CTRL, blue) and without indirect effect (ROCD, purple) and without BB emission (noBB, green) along with the flight altitude for OA on 19 March (b) The HALO flight and detailed locations on 19 March.

[Figure]

Figure R3 (a)-(c) Spatial distribution of control simulation for PM2.5 (color) and cloud water(dot-shaded) at altitude 2000 m at 02:00, 04:00, and 06:00 UTC, respectively, on 19 March 2018. (d)-(f): the effect of indirect effect, i.e. the simulation difference between the case CTRL (control) simulation and ROD (running only considered direct effect), respectively. (g)-(i): the effect of BB plume transport, i.e. the simulation difference between with and without BB emission, respectively.

[Figure]

Figure R4 Simulated vertical distribution of cloud water difference between with and without indirect effect in the model on 19 March 2018.

[Figure]

Figure R5 Simulated mean downward short wave flux (DSWF) (red) reduction at the ground surface over the regions in Fig.1a and contributed by BB (%, blue), aerosol indirect effect (%, dashed) during 15-19 March 2018. (a) ECSA (b) SCA

3. The concluding statements (L575-579) aren't necessarily based on the findings from this study.

R: Thank you for the suggestions. This paragraph has been amended in this revision.

Minor comments:

4. It'd be helpful to add a Table to list the WRF-Chem model configuration. Some of the settings are listed in the text. Information on the lateral boundary conditions for the chemical species, their cycling between the subsequent simulations and fire plumerise are missing. How the wet removal of the gas and aerosol species are parameterized in the model?

R: Thank you for the suggestion. A table listing the model configuration has been added in this revision (Table1, please see in Q2(a)). To avoid the global information disturbing the simulation result, we do not include global chemical information at the lateral boundary. We used the WRF-Chem (V4.1.1) default grid-scale wet-scavenging scheme, which is based on Neu and Prather (2012) and updated by Bela et al. (2016) to include ice retention factors in the grid-scale wet-scavenging. When wet scavenging occurs, the amount of trace gas that dissolves in cloud water is governed by Henry's law. The chemistry option, RACM chemistry with MADE/VBS aerosol scheme was used in this study (Table R1 in Q2a).

References:

Bela, M. M., Barth, M. C., Toon, O. B., Fried, A., Homeyer, C. R., Morrison, H., et al. (2016). Wet scavenging of soluble gases in DC3 deep convective storms using WRF-Chem simulations and aircraft observations. Journal of Geophysical Research: Atmospheres, 121,4233–4257. https://doi.org/10.1002/2015JD024623

Neu, J. L., & Prather, M. J. (2012). Toward a more physical representation of precipitation scavenging in global chemistry models: Cloud overlap and ice physics and their impact on tropospheric ozone. Atmospheric Chemistry and Physics, 12(7), 3289–3310. https://doi.org/10.5194/acp-12-3289-2012

5. The paper doesn't provide any information about the measurement uncertainties for the chemical species. For instance, the AMS data (OA, sulfate concentrations reported here) usually have significant uncertainty due to the collection efficiency and cutoff size (< 1 micron).

R: The uncertainty of the AMS regarding ionization efficiency (IE) and collection efficiency (CE) was determined to be 34% for ammonium and nitrate, 36% for sulfate, and 38% for organics. The related information can be seen in Bahreini et al.,( 2009; including the auxiliary material S1: https://agupubs.onlinelibrary.wiley.com/doi/full/10.1029/2008JD011493).

Middlebrook et al (2012) (http://dx.doi.org/10.1080/02786826.2011.620041) recommend a collection efficiency for low nitrate concentrations of 0.5, which is the value we used. Then the overall uncertainty is somewhat reduced and estimated to be around 30%.

The size cut of the inlet is not an uncertainty, but an instrument feature, therefore one should always refer to "submicron aerosol" when describing AMS data (at least those with a classical inlet setup).

**References:**

Bahreini R., Ervens, B., Middlebrook, A. M., et al., Organic aerosol formation in urban and industrial plumes near Houston and Dallas, Texas,J. Geophys. Res., 114, D00F16, doi:10.1029/2008JD011493, 2009.

Middlebrook A. M., Bahreini R., Jimenez Jose L. and Canagaratna M. R. ) Evaluation of Composition-Dependent Collection Efficiencies for the Aerodyne Aerosol Mass Spectrometer using Field Data, Aerosol Science and Technology, 46:3, 258-271,DOI: 10.1080/02786826.2011.620041, 2012.

6. L156: For WRF-Chem the more recent paper (Powers et al.) can be also cited.

   R: Thank you for the suggestions. The recent paper (Power et al. 2017), has been cited in this revision.   **(L160)**

7. L272: What do you mean by "stable"?

   R: The text has been dropped in this revision.

8. Chapter 3.3: this chapter needs to be shortened.

   R: This section has been modified in this revision.

Response to Reviewer #2

1. (a) Observational data to assess the regional impact of biomass burning plumes originating from Indochina peninsula have been still rare and thus the aircraft observations and the associated model simulations presented in this manuscript are important. However, more clarification is needed to justify some of the conclusions. First, the authors should be able to specify the locations and date of the fires likely affecting the studied events.

R: Figure R1 showed the fire hot spots during the study period between 14-19 and indicated a high dentist of fires frequently occurring during the study period (also, please see the annual fire number variations shown below in Q5). In this study, the biomass burning (BB) event day was on 19 March, and the backward trajectories in the East China Sea (ECS) indicated air masses mainly transported 48-72 and even 96 hrs (15-17 March) ago as shown in Figure R2. According to the backward trajectories in Figure R2, the locations of fire hot spots were distributed randomly in Indochina Peninsula (mainly 18-28 N, 90-110 E; i.e., Myanmar, Laos, Thailand, and Vietnam) as shown in Figure R1.

[Figure]

Figure R1    MODIS fire host sports during 14-19 March 2018.

[Figure]

Figure R2 Result of the HYSPLIT model backward trajectory analysis at 3000 meters with multiple points by 1˚X1˚ in the area (122-130˚E, 28-33 ˚N) of East China Sea started at 04 UTC 19 March 2018.

(b) Precipitation and cloud processes during the long-range transport should be mentioned even if negligible, to characterize potential loss of aerosol species and to fully attribute the differences to the emissions

R: Thank you for the comments. Yes, the precipitation and cloud processes should play a role in the loss of aerosols during long-range transport. Figure R3 showed the simulated difference between the control simulation and the aerosol indirect effect turned off on the daily accumulation rainfall between 00:00UTC 17 and 09:00 UTC 19 March 2018. We have found that light rainfall difference (< 1 mm/day) due to wet scavenging along the frontal system (Figure R3 d-f) during the study period. We further presented the simulation of aerosol species (OA, BC) along the aircraft for the simulation only considered direct effect (case ROCD ) as shown in Figure 9 in this revision. Most of the time the difference was not significant except for during 03:30-04:20 UTC along the flight where it was located north of 28 N and a frontal cloud band existed as shown in Fig.2g. In other words, the effect of wet scavenging reduced the aerosol concentration bias in the ECS and SC, as for the frontal system providing the moist air mass in the event flight F0319 **(L436-441)**. (please also see in Q16, Figure R6 in this response)

[Figure]

R3 The simulated daily accumulation rainfall (a-c) and the difference between the wet scavenging turned on and off (d-f) during 17-19 March, 2018.

(c) The degrees of overestimation should be quantitatively assessed and mentioned in the abstract more clearly.
R: The statement about this part has been included in this revision **(L56-58)**

**2.** Second, details of chemical pathways that enhanced the OH and HO2 levels in the model should be described. This part is purely from model results - to provide associated observational evidence is recommended (also for J values, cloud condensation nuclei, and cloud water).

R: In the RACM model (Stockwell et al. 1997), the production of OH during the day was dominated by $O(^1D) + H_2O$ and $HO_2 + NO$ while $HO_2 + NO$ was dominant in the early morning and late afternoon (Kanaya et al. 2001). In polluted environments, the photolysis of other oxygen-containing species, such as nitrous acid (HONO), formaldehyde (HCHO), and hydrogen peroxide ($H_2O_2$) can also be important $HO_x$ sources (Seinfeld and Pandis, 2006). In the troposphere, many gas phase species such as VOC, CO, $NO_2$, and other species are mainly removed through their reaction with OH (sink). Loss of $HO_2$ was dominated by its reaction with NO and O. Furthermore, the reactions of $HO_2$ with $RO_2$, $HO_2$, and $O_3$ also played a role ( Kanaya et al. 2001; Seinfeld and Pandis, 2006)

Figure R4 indicated the aircraft measurement for the J value (JO$^1$D) and CCN (Cloud Condensation Nuclei; at a constant instrument supersaturation of 0.38 %) along the flight on 19 March, 2018. The CCN number concentration (per cm$^3$), was consistently increased with the aerosol species (such as OA) as the flight passed through the BPTL (2000-4000 m). J value (black solid line) was high as the flight above 4000 m (03-05 UTC). In the contrast, J value decreased as the flight below the BPTL (e.g

[Figure]

before 01:00 UTC and after 05 UTC). **(L361-366)** (also see Supplementary S3)

Figure R4 Observed OA concentration (green), J value (O$^1$D)(black solid) and CCN number (cm$^{-3}$)(red) along with the flight altitude (dot) on 19 March 2018.

**References:**

Kanaya Y., Y. Sadanaga, K. Nakamura and H. Akimoto, J. Geophys. Res., Behavior of OH and HO2 radicals during the observations at a remote island of Okinawa (ORINO99) field campaign[Atmos.], 2001, 106, 24209-24223.

Seinfeld, J., and S. Pandis (2006), *Atmospheric Chemistry and Physics: From Air Pollution to Climate Change*, Wiley, New York.

Specific comments.
3. Line 101. Similar to what?
R: Thank you for the comments. This sentence has been amended in this revision. We originally intended to present the emission uncertainties existing pointed out by Shi and Yamaguchi (2014). **(L104-106)**

4. Line 154. The authors state that OH and HO2 are listed in the HALO aircraft data but in fact they were not observed. (only HO2+RO2)
R: Thank you for the comment. Yes, the individual OH or HO$_2$ is not available in HALO measurement. It is only for modeling output. The HALO data, the total sum of peroxy radicals (RO2*= HO2 + ΣRO2,where R stands for any organic chain) has been measured

by the PeRCEAS (Peroxy Radical Chemical Enhancement and Absorption Spectrometer) instrument. The text has been amended in this revision **(L157)**. For the related response, please see in Q2.

5. Line 178. Is acetone dominant for KET? For example, MACR and MVK from isoprene chemistry could also contribute? Emissions of acetone from anthropogenic and biomass burning should be briefly discussed.

R: In RACM, the "KET" represents acetone and higher saturated ketones (KET). A common technique for reducing the number of VOC species carried in a chemical mechanism is to lump VOCs, molecule by molecule, into surrogate species representing whole classes of compounds with similar structures (Stockwell et al., 2011). The chemistry of KET is treated as a mixture of 50% acetone and 50% methyl ethyl ketone (by mole fraction)( Stockwell et al., 1997). According to Singh et al. (1994), secondary formation from the atmospheric oxidation of precursor hydrocarbons (principally propane, isobutane, and isobutene) provides the single largest source (51%) of acetone. The remainder is attributable to biomass burning (26%), direct biogenic missions( 21%), and primary anthropogenic emissions( 3%). **(L181-184)**

Also, there is a species "MACR" which represents methacrolein and other unsaturated monoaldehydes in RACM. The species "MVK" is not included in the RACM.

References:

Singh H.B., Hara D.O., Herlth D., Sachse W., Blake D.R., Bradshaw J.D., Kanakidou M., Crutzen P. J., Acetone in the atmosphere: Distribution, sources, and sink., J. Geophys. Res., 99,1805-1819, 1994.

Stockwell, W. R., Lawson, C. V., Saunders, E., and Goliff, W. S.: A review of tropospheric atmospheric chemistry and gas-phase chemical mechanisms for air quality modeling, Atmosphere, 3, 1–32, doi:10.3390/atmos3010001,. 879, 880, 2011

Stockwell, W. R., Kirchner F., Kuhn M.: A new mechanism for regional atmospheric chemistry modeling, J. Geophys. Res., 102, 25847–25879, 1997.

6. Lines 184, 190, and 412. MICS-Asia III and TEDS emissions were used - for which year?

R: The emission for MICS-Asia III was in 2010 and TEDS version for this study was 2013 (as shown in the original version Line 193). **(L190-192; 198)**

7. Line 200. Can the authors describe whether the intensity of biomass burning in Indochina peninsula during this particular period in 2018 was at normal level or not, in comparison to other years?

R: Figure R5 showed the statistic of annual variations of active fire detections from Terra MOIDS satellite in spring (March, April, and May) over Indochina (10˚N to 25 ˚N, 90 ˚E to 110 ˚E) from 2011 to 2020. Each MODIS active fire/thermal hotspot location represents the center of a 1km pixel that is flagged by the algorithm as containing one or more fires within the pixel. Combined (Terra and Aqua) MODIS NRT active fire products (MCD14DL) are processed using the standard MOD14/MYD14 Fire and Thermal Anomalies algorithm (https://www.earthdata.nasa.gov/learn/find-data/near-real-time/firms). Data showed the year 2018 and 2017 were 33% (1/3) lower than other years.

[Figure]

Figure R5: The annual variations of active fire detections from Terra MOIDS satellite in spring (March, April, and May) over Indochina (10˚N to 25 ˚N, 90 ˚E to 110 ˚E) from 2011 to 2020

8. Line 203. It seems that the center of the high pressure system is present over the Japan (Japan Sea), rather than Korea.

R: Text has been amended in this revision. **(L208-209)**

9. Line 269. SO2 enhancement is attributed to Japan - perhaps volcanoes have contributed?

R: During our EMeRGe-Asia campaign period, there was a volcano namely " Kirishimayama" eruption that started on 01 March and stopped on 27 June, 2018. (https://volcano.si.edu/faq/index.cfm?question=eruptionsbyyear&checkyear=2018).

Our flight to the north was between 30-32 N (manuscript Figure 6a), i.e. in the south of Japan on 19 March, 2018. In other words, it can not rule out the contribution of this

volcano to this high sulfate concentration under favorable weather conditions. However, it is difficult to identify due to unregular release from the volcano and weather conditions.

10. Line 311. Carmichael

R: The text has been amended in this revision.(L316)

11. Line 312. Figure 6b indicates biomass burning influence is spread to the north of 30 degN.

R: Thank you for the comments. Text has been amended in this revision. (L315-321) However, the ACN still could be around 300ppt or less as the flight at the north of 30 °N (during 3:30-4:30 UTC) and could be recognized as the contribution of BB (Förster et al. 2022). In other words, it might still have BB products being transported to the north of 30 °N under favorable weather conditions although the ACN concentration was low compared to the south of it at the layer of BPTL(between 2000 and 4000 m).

**Ref:**

Förster, E., Bönisch, H., Neumaier, M., Obersteiner, F., Zahn, A., Hilboll, A., Kalisz Hedegaard, A. B., Daskalakis, N., Poulidis, A. P., Vrekoussis, M., Lichtenstern, M., and Braesicke, P.: Chemical and dynamical identification of emission outflows during the HALO campaign EMeRGe in Europe and Asia, Atmos. Chem. Phys. Discuss. [preprint], https://doi.org/10.5194/acp-2022-455, in review, 2022.

12. Line 338. As ACN and ACE contain oxygen and nitrogen in their molecules, they are not hydrocarbons.

R: The text has been amended in this revision. (L347)

13. Line 350 and 352. Use uppercase 1 for J(O1D).

R: Thanks for the comments. The text has been amended in this revision. (L358, 361)

14. Lines 351 and 479. Whether aerosols increase or decrease $J(O1D)$ will be dependent on the assumed single-scattering albedo. Any evidence from direct observations of the J values?

R: Please see the response in Q2 and Figure R4.

15. Line 404. It is better to confirm that the CO hemispheric baseline is not overestimated.

R: Time series of background CO mixing ratios during the 1990s, averaged over the

Extratropical Northern Hemisphere was between 100-200 ppb (Wotawa et al. 2001). Furthermore, our study focused on the BB contribution during the event day (19 March 2018) which is a short-term period as presented in this study. The simulated CO concentration difference with and without BB emission is not related to the global information.

Ref: Wotawa G., Novelli P.C., Trainer M., Granier C.: Inter-annual variability of summertime CO concentrations in the Northern Hemisphere explained by boreal forest fires in North America and Russia: Geophys. Res., Lett., 28, 4575-4578, 2001.

16. Line 416. It is important to confirm that OA and BC have not been removed by wet deposition on the way of transport, to better attribute the model's overestimation to emissions.

R: Thank you for the suggestions. We further checked and identified that OA and BC have been removed by wet scavenging (due to aerosol indirect effect) during 03:30-04:20 UTC. In other words, the effect of wet scavenging reduced the aerosol concentration bias in the ECS and SC, as for the frontal system providing the moist air mass in the event flight F0319. When the simulation only considered the direct effect (case ROCD, purple) below,the overestimations were increased as shown in Figure 9b-c. **(L436-441)**

[Figure]

Figure R6 Observed (OBS, red) and simulated concentration (CTRL, blue), and the simulation without indirect effect (ROCD, purple), without BB emission (noBB, green) along with the flight altitude for (a) OA ($\mu$g m$^{-3}$) (b) BC ($\mu$g m$^{-3}$) on 19 March 2018.

17. Line 447. "detraining" is difficult to understand.

R:    Thank you for the comments. The text has been dropped in this revision.

18. Line 457. The sentence starting with" The variation trend of PM2.5   ..." needs to be rewritten.

R: Thank you for the comments. This sentence has been rewritten in this revision. **(L476-478)**

19. Line 471. Which processes were responsible for the OH and HO2 enhancement? How well VOCs emissions and chemistry were treated to describe the OH and HO2 budget?

R: In general, the photolysis of ozone followed by the subsequent reaction of $O(^1D)$ with water vapor is the main HOx source during daytime in the clean troposphere. In the troposphere, many gas phase species such as VOC, CO, NO2 and other species are mainly removed through their reaction with OH (sink). HO2 production followed the reaction of peroxy radicals with NO, with additional contributions from formaldehyde photolysis and reactions of OH with CO and formaldehyde. Please also see the response in Q2.

20. Line 513. Any observational evidence of CCN or cloud water enhancement, attributable to the biomass burning plume?

R: This question has been responded in Q2 and Figure R4.

21. Figure 3a, b: As the highest CO area is distant from Indochina peninsula on the day, the authors should be able to state the possible locations and date of fires producing the plumes.

R:    To identify the high CO concentration in the South China Sea at 1000 meters in Figures 3a and b, the backward trajectories with multiple points by 1°X1° in the area (110-115°E, 17.5-22.5 °N) in the South China Sea started at 00 UTC 17 March 2018 as shown in Figure R7.   The locations of fire hot spots were distributed randomly in Indochina Peninsula as shown in Figure R1 ( Q1). The backward trajectories in the South China Sea indicated air masses mainly transported 48-72 and even 96 hrs. In other words, there could be contributed by fires occurring between 100-110 E and 12-20 N (Myanmar, Laos, Thailand, and Vietnam) during 13-15 March 2018.

[Figure]

Figure R7 Result of the HYSPLIT model backward trajectory analysis at 1000 meters with multiple points by 1˚X1˚ in the area (110-115˚E, 17.5-22.5 ˚N) of East China Sea started at 00 UTC 17 March 2018

---

## Author Response (AR2)

In general the revised manuscript has been improved accordingly. However, the authors are recommended to consider the following points for further improvement.

R: We greatly appreciate the reviewer's constructive comments and, accordingly, have completed the revision and the responses as following:

1. I find several places where the authors' reply contents should be included in the text for clarification.

First, the locations and date of the forest fires affecting the observed air masses should be mentioned in the text, as analyzed with Figure R7 (as between 100-110 E and 12-20 N (Myanmar, Laos, Thailand, and Vietnam) during 13-15 March 2018).

R: Thank you for the suggestion. Figure R7 has been included in this revision to further explain the locations and dates of the forest fires affecting the observed air masses. (Figures 3e and 4e in this revision; **Line 243-251** and **Line 260-263**)

Second, about the likely negligible effect from wet deposition on the loss of BC and OA during the transport to secure the conclusion that the OA and BC emissions were likely overestimated.

R: Thank you for this suggestion. This point has been included in this revision **(Line 452-457 )**

Third, about the relatively weaker forest fire activity in the year of 2018 over Indochina (Figure R5).

R: Figure R5 also has been included in the text in this revision (Supplementary Figure S1a; **Line 207-209**).

2. The statements about OH and HO2 in the paragraph starting from line 490 (ATC1.pdf) need justification. First, the term "observed" in line 492 needs to be avoided as HO2 is only modeled.

R: Thank you for the comment. The text "observed" has been dropped in this revision.

Second, I disagree with the statement in line 496 that trace constituents from BB were expected to increase OH and HO2, as the emitted VOCs could reduce OH (depending on the chemical mechanism).I am afraid that most light-weighted NMHCs were unmeasured and thus were unconstrained in the model - such overstatement about the behavior of HOx radicals should be avoided.

R: We agree with the concern point from this reviewer. The statement in line 496 has been dropped in this revision.